


# Analysis of regional $CO_2$ contributions at the high Alpine observatory Jungfraujoch by means of atmospheric transport simulations and $\delta^{13}C$

Simone M. Pieber[1], Béla Tuzson[1], Stephan Henne[1], Ute Karstens[2], Christoph Gerbig[3], Frank-Thomas Koch[3,4], Dominik Brunner[1], Martin Steinbacher[1] and Lukas Emmenegger[1]

[1] Laboratory for Air Pollution and Environmental Technology, Empa, Switzerland
[2] ICOS Carbon Portal, Lund University, Sweden
[3] Max Planck Institute (MPI) for Biogeochemistry (BGC), Jena, Germany
[4] Meteorological Observatory Hohenpeissenberg, Deutscher Wetterdienst, Germany

simone.pieber@empa.ch

**Abstract.** Understanding of regional greenhouse gas emissions into the atmosphere is a prerequisite to mitigate climate change. In this study, we investigated the regional contributions of carbon dioxide ($CO_2$) at the location of the high Alpine observatory Jungfraujoch ("JFJ", Switzerland, 3580 m a.s.l.). To this purpose, we combined receptor-oriented atmospheric transport simulations for $CO_2$ concentration in the period of 2009–2017 with stable carbon isotope ($\delta^{13}C$-$CO_2$) information. We applied two Lagrangian particle dispersion models driven by output from two different numerical weather prediction systems (FLEXPART-COSMO and STILT-ECMWF) in order to simulate $CO_2$ concentration at JFJ based on regional $CO_2$ fluxes, to estimate atmospheric $\delta^{13}C$-$CO_2$, and to obtain model-based estimates of the mixed source signatures ($\delta^{13}C_m$). Anthropogenic fluxes were taken from a fuel type-specific version of the EDGAR v4.3 inventory and ecosystem fluxes were based on the Vegetation Photosynthesis and Respiration Model (VPRM). The simulations of $CO_2$, $\delta^{13}C$-$CO_2$ and $\delta^{13}C_m$ were then compared to observations performed by quantum cascade laser absorption spectroscopy. Around 40 % of the regional $CO_2$ variability above or below the large-scale background was captured by the models, and up to 35 % of the regional variability in $\delta^{13}C$-$CO_2$. This is remarkable considering the complex Alpine topography, the low intensity of regional signals at JFJ, and the challenging measurements. Best agreement between simulations and observations in terms of short-term variability and intensity of the signals for $CO_2$ and $\delta^{13}C$-$CO_2$ was found between late autumn and early spring. The agreement was inferior in the early autumn periods and during summer. This may be associated with the atmospheric transport representation in the models. In addition, the net ecosystem exchange fluxes are a possible source of error, either through inaccuracies in their representation in VPRM for the (Alpine) vegetation or through a day (uptake) vs. night (respiration) transport discrimination to JFJ. Furthermore, the simulations suggest that JFJ is subject to relatively small regional anthropogenic contributions, due to its remote location (elevated and far from major anthropogenic sources), and the limited planetary boundary layer-influence during winter. Instead, the station is primarily exposed to summer-time ecosystem $CO_2$ contributions, which are dominated by rather nearby sources (within 100 km). Even during winter, simulated gross ecosystem respiration accounted for approximately 50 % of all contributions to the $CO_2$ concentrations above the largescale background. The model-based monthly mean $\delta^{13}C_m$ ranged from –22 ‰ in winter to –28 ‰ in summer and reached the most depleted values of –35 ‰ at higher fractions of natural gas combustion, and the most enriched values of –17 to –12 ‰ when impacted by cement production emissions. Observation-based $\delta^{13}C_m$ values derived by a moving Keeling-plot approach were in good agreement with the model-based estimates. They exhibited a larger scatter, while model-based estimates spread in a more narrow range. Overall, observation-based $\delta^{13}C_m$ were limited to a smaller number of data points compared to model-based estimates owing to the stringent analysis prerequisites in combination with the low regional signal at JFJ.



## 1. Introduction

Reliable regional quantification of greenhouse gas (GHG) emissions into the atmosphere is a prerequisite to determine the effectiveness of mitigation strategies to limit global warming. Carbon dioxide ($CO_2$) is the prime player in these regards. Its atmospheric concentrations are altered by both anthropogenic and natural (terrestrial ecosystem and oceanic) fluxes (Friedlingstein et al., 2020). Remote sites are ideal to study large-scale and global emissions, but make it more challenging to characterize individual sources and sinks as during transport of air masses to remote sites the signals of individual sources and sinks become diluted and mixed. Thus, remote atmospheric sites typically focus on long-term trends, and, therefore, sporadic events are often discarded in the time series analyses. This leads to loss of potentially insightful information.

In this study, we focus on the information contained in the regional scale signals at the remote high altitude observatory Jungfraujoch (JFJ), situated in the Swiss Alps. Owing to its particular location in central Western Europe and its altitude of 3580 m above sea level (a.s.l.), it allows for studying background concentrations of air pollutants and GHGs in the lower free troposphere (Herrmann et al., 2015). These background conditions are representative of large spatial or temporal scale variations and not influenced by regional sources or sinks. Furthermore, regional signals transported from different regions within Western Europe and beyond reach the monitoring station intermittently (Henne et al., 2010). Thus JFJ offers both aspects: i) insight into the atmospheric background, and ii) an opportunity for studying GHGs and pollutants sources and sinks in the planetary boundary layer (PBL) on a regional scale. The latter is challenged, however, by low signal-to-background ratios, and requires high-precision instrumentation. In comparison to a typical low altitude site, the regional signal measured at JFJ is integrated over a larger concentration footprint (source area). This allows for a greater coverage per measurement, but also leads to a higher degree of mixing of various sources and sinks. Atmospheric backward transport simulations can provide information about the history (location backward in time) of the sampled air mass and a quantitative relationship between atmospheric concentrations and sources or sinks (source/sink-receptor relationships) to combat this challenge. Although atmospheric transport and concentration simulations are particularly demanding for complex topography, observations at JFJ have been successfully combined with high-resolution transport simulations in previous inverse modelling studies to allocate and quantify emissions of $CH_4$ (Henne et al., 2016) and halocarbons (Keller et al., 2011; Brunner et al., 2017; Vollmer et al., 2021).

The same task, however, is more challenging for $CO_2$, because of the strong contribution of natural processes in addition to anthropogenic sources, the interplay between signals from sources and sinks, and the large temporal variability and broad distribution, especially of the natural fluxes. In this case, multi-tracer approaches are favourable that allow for separation of different processes based on composition characteristics. For instance, carbon monoxide (CO), which is co-emitted during combustion processes, was used to identify combustion-related and ecosystem contributions to the observed $CO_2$ signals (, Levin and Karstens (2007), Vogel et al. (2010), Vardag et al. (2015) or Oney et al. (2017)). However, this method suffers from variable $CO/CO_2$ emission ratios and atmospheric production and loss of CO. The approach is most promising when all sources/sinks in the footprint area are well characterised, yet remains challenging for sites with low signal-to-background ratios, such as JFJ.



Other promising tracers are isotopes, as isotope composition measurements can provide valuable information on the sources and sinks contributing to the regional signal. Today, sufficiently precise instrumentation is available that allows to measure the stable isotope composition at high precision and temporal resolution for several natural GHGs, see Tuzson et al. (2008b) for $CO_2$, Eyer et al. (2016) for $CH_4$ and Waechter et al. (2008) for

$N_2O$. Applying these or similar techniques, for instance, Röckmann et al. (2016), Hoheisel et al. (2019), Menoud et al. (2020), Xueref-Remy et al. (2020) and Zazzeri et al. (2015 and 2017) derived observation-based isotope source signature estimates from measurements conducted to study near-source or regional-scale $CH_4$ plumes. Harris et al. (2017a and 2017b) and Yu et al. (2020) presented similar analyses for $N_2O$. These studies took advantage of double-isotope constraints, i.e., $\delta^{13}C$-$CH_4$ and $\delta^2H$-$CH_4$ for $CH_4$, and $\delta^{15}N$-$N_2O$ and $\delta^{18}O$-$N_2O$ for $N_2O$ and provided very

promising results, although the availability of long-term data sets is still very limited.

The stable carbon isotope of $CO_2$, $\delta^{13}C$-$CO_2$, is an attractive tracer for $CO_2$ sources and sinks. So far it has been largely employed for analysis of long-term atmospheric background trends (Keeling et al., 1979; Graven et al., 2017), in global ecosystem studies (Ballantyne et al., 2011; Keeling et al., 2017; Van Der Velde et al., 2018), as well as to characterise emissions close to a source. Traditionally, the near-source $\delta^{13}C$-$CO_2$ studies focus on

ecosystem processes in areas with limited anthropogenic influence (Pataki et al., 2003), or on anthropogenic emissions under limited ecosystem influences, such as the vehicle tunnel study by Popa et al. (2014). However, the current instrumental capability of high precision $\delta^{13}C$-$CO_2$ observations at high temporal resolution (e.g., Sturm et al. (2013) or Vogel et al. (2013)) opens up new opportunities to disentangle $CO_2$ in a more complex setting. For instance, Pugliese et al. (2017) and Vardag et al. (2016) recently studied urban air masses, and Ghasemifard et al.

(2019) and Tuzson et al. (2011) attempted to characterise specific regional scale $CO_2$ signals at remote sites. These studies used hourly to daily resolution, and compared observation-based (mixed) isotope source signatures ($\delta^{13}C_m$) with literature information on source-specific signatures ($\delta^{13}C_s$); often, however, reducing the data to few particular pollution events, as this method is applicable only under very stringent conditions (see e.g., Zobitz et al., 2006).

These source identification or apportionment studies use $\delta^{13}C_s$ to discriminate $CO_2$ emissions from fuel

burning; in particular to distinguish gaseous (–40 ‰ for thermogenesis gas,–60 ‰ for microbial gas) from solid (–20 ‰ to –25 ‰, for wood/coal) or liquid fuels (–25 ‰ to –32 ‰, for heating oil, gasoline and diesel). (All values are based on Andres et al. (1994), Vardag et al. (2015 and 2016) and Sherwood et al. (2017), and presented based on the Vienna Pee Dee Belemnite (VPDB) reference scale.) However, ecosystem processes and their $\delta^{13}C_s$ add further complexity, as they are highly dependent on plant growth conditions (ambient humidity, $CO_2$ concentration)

and photosynthetic pathway (C3- vs C4-plants), detailed by Hare et al. (2018) and Kohn (2010). $CO_2$ from C3 plants carries a mean respiration signature of –27.5 ‰ with a range from –20 ‰ to –37 ‰ under arid, respectively humid, conditions. The smallest $^{13}C$ uptake relative to $^{12}C$, i.e. highest fractionation and thus the most depleted $\delta^{13}C_s$ of –37 ‰, is observed in tropical forests, and of little relevance for European ecosystems. C4-plants exhibit distinctly smaller $^{13}C$ fractionation during photosynthesis and can be distinguished from C3 plants based on their peculiar

$\delta^{13}C_s$ of about –12.5 ‰. In Europe, C4 plants are mainly present in croplands owing to extensive maize production. Overall, however, C3 plants, whose $\delta^{13}C_s$ overlap with anthropogenic sources, dominate the European and global ecosystems (Ballantyne et al., 2011). Thus, the $\delta^{13}C_s$ approach proves particularly meaningful among either the anthropogenic or the ecosystem carbon pool itself.



The stable oxygen isotope ratio of $CO_2$, $\delta^{18}O$-$CO_2$, is, aside of the carbon cycle, subject to the global water

cycle (e.g., Welp et al., 2011) due to the isotope exchange between water and $CO_2$ and thus ambiguous as $CO_2$ tracer. However, the radiocarbon signature may be used to quantify fossil fuel contributions to atmospheric $CO_2$, as done by e.g., Levin et al. (2003), Vogel et al. (2010), Turnbull et al. (2015), Berhanu et al. (2017), or Wenger et al. (2019). The $\Delta^{14}C$ allows primarily for discrimination of fossil versus ecosystem carbon. Once this is accomplished, $\delta^{13}C$ provides further insight into the partitioning of fuel types among the fossil pool, or of contributions from

different photosynthetic pathways among the ecosystem pool. Such dual carbon-isotope approaches making use of co-located $\delta^{13}C$ and $\Delta^{14}C$ measurements have already proven successful for carbon source apportionment in few gas- (Meijer et al., 1996; Zondervan and Meijer, 1996) and particle phase studies (Winiger et al., 2019; Andersson et al., 2015). Yet, studies are currently limited to infrequent sampling at few locations, since the involved laboratory analyses are costly, and high frequency, in-situ measurement techniques with sufficient precision for atmospheric

$\Delta^{14}C$-$CO_2$ currently unavailable, despite first developments (e.g., Genoud et al., 2019; Galli et al., 2011).

Despite these promising multi-tracer ($CO_2$, $CO$) and multi-isotope ($\delta^{13}C$ and $\Delta^{14}C$) approaches, the low signal-to-background ratios at remote sites still remain a challenge as highlighted by Vardag et al. (2015). Thus, combining measurements in addition with atmospheric simulations is essential for regional $CO_2$ apportionment. Yet, to date, few studies have performed hourly-scale regional simulations of $CO_2$ concentration and/or provide

"model-based" atmospheric $\delta^{13}C$-$CO_2$ or mixed isotope source signatures ($\delta^{13}C_m$) for a comparison with observations. The available studies currently include two ground-based urban locations (Pugliese-Domenikos et al. (2019) and Vardag et al. (2016)), and one rural tall tower location (Wenger et al., 2019).

Here, we address the situation at the high Alpine observatory JFJ. We aim at challenging our understanding of the contribution of $CO_2$ sources and sinks within the European domain to the regional $CO_2$ concentration

variability at JFJ, and at evaluating model-based $\delta^{13}C$-$CO_2$ and model-based mixed isotope source signatures ($\delta^{13}C_m$) against observations. To this end, we employ long-term regional $CO_2$ simulations for JFJ for a nine-year period (2009-2017) at 3-hourly time-resolution, using two different atmospheric transport models. We compare the model-based data to atmospheric observations, making use of the unique long-term high-frequency observations of $CO_2$ and $\delta^{13}C$-$CO_2$ measured by quantum cascade laser absorption spectroscopy (QCLAS) since 2008 (Sturm et al.,

2013; Tuzson et al., 2011), and deploy a moving Keeling plot method to obtain observation-based $\delta^{13}C_m$.

## 2. Methods

### 2.1 Site description

The High Altitude Research Station Jungfraujoch (JFJ) is located at 7°59'20'' E, 46°32'53'' N in the Swiss Alps,

at an altitude of 3580 m a.s.l. on a mountain saddle between the peaks of Jungfrau and Mönch (both > 4000 m a.s.l.). As part of the Swiss long-term national monitoring network (NABEL), regular measurements of air pollutants and GHGs are performed at JFJ since the 1970s (Buchmann et al., 2016). The station contributes to European (EMEP) and global (Global Atmospheric Watch; GAW) monitoring programmes and was labelled as class 1 station within the European Integrated Carbon Observing System (ICOS) in 2018 (Yver-Kwok et al., 2020).



## 2.2 Atmospheric Transport Simulations

Atmospheric $CO_2$ concentration simulations were conducted for the period 2009-2017 with two distinct combinations of Lagrangian particle dispersion models (LPDM), meteorological input fields, domain size and spatial resolution (Table 1). Both models were run in a receptor-oriented approach, following 'sampled' air masses backward in time, and as such providing surface source sensitivities ("footprints"). Convoluting these with spatially and temporally resolved $CO_2$ fluxes allows for quantitative simulations of $CO_2$ concentrations at the receptor site (Seibert and Frank, 2004). Here, we use the fuel type-specific version of the Emissions Database for Global Atmospheric Research (EDGAR v4.3) inventory and the Vegetation Photosynthesis and Respiration Model (VPRM) to account for anthropogenic and ecosystem $CO_2$ fluxes, respectively. The simulated $CO_2$ mixing ratios are reported in ppm, and we refer to them as "concentration" for readability. In order to disentangle the influence of the underlying $CO_2$ fluxes and the transport dynamics on the simulated $CO_2$ concentrations at JFJ, the influence of various parameters such as the domain size, the meteorological input fields, or the LPDM implementation was investigated in dedicated simulations with synthetic $CO_2$ fluxes in Appendix A1.

**Table 1.** Overview of atmospheric transport simulation models and their associated parameters.

| LPDM | Meteo. input | Approximate spatial resolution (km²) | Domain* | Integration period (days) | Release height (m asl) | Sampling height (m) | Temporal resolution | $CO_2$ fluxes |
|---|---|---|---|---|---|---|---|---|
| FLEX-PART | MeteoSwiss COSMO | $7 \times 7$ | WEU | 4 | 3100 | 50 | 3-hourly avg. | EDGAR v4.3 (pre-release), VPRM offline (Gerbig and Koch, 2021) |
| STILT | ECMWF IFS | $25 \times 25$ $(10 \times 10)$ | EU | 10 | 3100 | $0.5 \times h_{PBL}$ | 3-hourly snapshots | EDGAR v4.3 (pre-release) VPRM online (Gerbig, 2021) |

\* "EU" and "WEU" refers to 33°N-73°N, -15-35°E, and 36.06-57.42°N, −11.92-21.04°E, respectively

### 2.2.1 FLEXPART-COSMO

A version of the LPDM FLEXPART (Pisso et al., 2019; Stohl et al., 2005) coupled to output from the regional numerical weather prediction model COSMO (Baldauf et al., 2011) was operated using operational analysis fields generated by MeteoSwiss (see Henne et al., 2016). The model was run in backward mode to calculate source sensitivities for JFJ. Within each 3-hourly interval, 50'000 model particles were initialized continuously at the receptor location and traced back in time for 4 days or until they left the model domain. FLEXPART considers transport by the mean atmospheric flow as well as turbulent and sub-grid scale convective mixing. COSMO analyses were available hourly at a horizontal resolution of approx. 7 km × 7 km over Western Europe (COSMO-7; 36.06 – 57.42°N, −11.92 – 21.04°E; Figure S1). The horizontal resolution of the model does not resolve the steep topography around JFJ. Hence, a difference between observatory and model altitude exists. In previous studies (e.g., Keller et al., 2011), the optimal release height was determined to be around 3100 m above sea level when using COSMO-7 inputs, which is between the true altitude (3580 m) and the model topography (2650 m) at JFJ. Surface source sensitivities were determined from the location of model particles below a sampling height of 50 m and stored 3-hourly along the backward simulation, allowing for a 3-hourly coupling to temporally variable surface fluxes.



### 2.2.2 STILT-ECMWF

The Stochastic Time Inverted Lagrangian Transport (STILT) Model, first described by Lin et al. (2003), was driven by the numerical weather forecast fields from the European Centre for Medium-Range Weather Forecasts (ECMWF), as previously presented by Trusilova et al. (2010) and Kountouris et al. (2018a). The simulations for

JFJ were performed at the same release height as with FLEXPART-COSMO (3100 m.a.l.), corresponding to 960 m above the model topography. STILT-ECMWF simulations are also routinely performed within the activities of the ICOS Carbon Portal (CP), albeit at a release height of 720 m above model ground (2860 m a.s.l.) for the default products for JFJ (https://stilt.icos-cp.eu/worker/). The particles are released instantly on a 3-hourly interval and traced back in time for 10 days or until they leave the European domain (33°N–73°N, 15°W–35°E, Figure S1). The

STILT calculations were driven by 3-hourly operational ECMWF-IFS analysis/forecast fields available at a resolution of 0.25° × 0.25° (approx. 25 km × 25 km), whereas STILT output was generated on a finer grid (approx. 10 km × 10 km). Surface source sensitivities were evaluated by using a variable sampling height ($0.5 \times h_{PBL}$), where $h_{PBL}$ is the PBL height diagnosed within STILT.

## 2.3 CO$_2$ fluxes and boundary conditions for the atmospheric transport simulations

### 2.3.1. Regional CO$_2$

#### *A) Anthropogenic Emissions*

Regional anthropogenic CO$_2$ concentrations for JFJ (CO$_2$.anthr) were calculated using emission fluxes based on a pre-release of EDGAR v4.3 (pers. comm. with G. Janssens-Maenhout). The inventory was disaggregated into fuel-type specific categories (Table S1), and provides annual emissions on a 0.1° × 0.1° grid (~10 km × 10 km) (Janssens-

Maenhout et al., 2019; Karstens, 2019). Here, we use 14 categories, representing 11 different fossil and biogenic fuel types as well as 3 non-fuel categories from cement and other production processes (Table 2). The CO$_2$.anthr comprises CO$_2$ from fuel-burning CO$_2$ (oil, gas, coal, liquid biofuels, biogas, solid biomass), and CO$_2$ from cement and other industrial production (referred to as CO$_2$.cement collectively). We temporally extrapolated the inventory, which was established for the base year 2010, using annual scaling factors per country and category based on data

from BP (bp, 2019), see Table S2. Additionally, we applied seasonal, weekly, and diurnal time factors for different anthropogenic categories. These are based on MACC-TNO (Kuenen et al., 2014) and available in Table S3.

#### *B) Ecosystem Fluxes*

Regional ecosystem CO$_2$ fluxes were based on the VPRM (Mahadevan et al., 2008). Underlying parameters are specific for seven vegetation types (VT) including: 1) evergreen forest, 2) deciduous forest, 3) mixed forest, 4)

shrubland, 5) savanna, 6) cropland, 7) grassland. The VTs are based on the settings typically used within the ICOS Carbon Portal, although, for instance, category 5 (savanna) is irrelevant within the domain boundaries used for JFJ. An additional category "others" includes primarily water bodies and urban spaces for which VPRM does not estimate CO$_2$ fluxes and, hence, was excluded from the final analysis. The VT maps underlying VPRM are based on the synergetic land cover product (SYNMAP, Jung et al., 2006). A map showing the dominant category per grid

as used in our study is provided in Figure S2. Note that oceanic sources and sinks (including oceanic biomass), as well as human or animal respiration (see e.g., Ciais et al., 2020) and wildfire related emissions were not included,





and are expected to be a minor contribution to the regional signal at JFJ. With FLEXPART-COSMO, we use an offline version of VPRM (Gerbig and Koch, 2021) based on the same ECMWF meteorological analysis as in STILT-ECMWF. Although the fluxes are generated based on the individual VTs, ecosystem respiration ($CO_2$.resp),

ecosystem uptake (also referred to as gross ecosystem exchange, and thus abbreviated $CO_2$.gee), and net ecosystem exchange ($CO_2$.nee = $CO_2$.gee – $CO_2$.resp) are provided only as a total over all VTs. The STILT-ECMWF is coupled online with VPRM and allows extracting $CO_2$ concentration contributions at JFJ for $CO_2$.nee, $CO_2$.gee and $CO_2$.resp for the individual VTs separately. The *online* VPRM parametrisation initially presented by Kountouris et al. (2018b) was updated for our study (Gerbig, 2021). A dedicated evaluation of the online compared to the offline

implementation with STILT-ECMWF for at JFJ yielded comparable results for $CO_2$.nee, $CO_2$.gee and $CO_2$.resp.

**2.3.2 Background $CO_2$**

We use the Jena CarboScope (JCS) global atmospheric $CO_2$ product for the determination of the $CO_2$ boundary conditions. These simulations are based on optimized fluxes (Rödenbeck, 2005) and available at http://www.bgc-jena.mpg.de/CarboScope/. We used three-dimensional CarboScope fields (version/experiment: s04oc_v4.3) with a

temporal resolution of 6 hours and interpolated concentrations in space and time to the endpoints of model particles. The mean over all model particles of a given release forms the background concentration (denoted $f_b$ herein) at the time of the release. We observed a higher short-term variability in the simulated background $CO_2$ concentration for FLEXPART-COSMO compared to STILT-ECMWF, which is a consequence of the smaller domain size, in particular towards Eastern Europe, and shorter backward-integration time (4 days versus 10 days).

**2.3.3 Total $CO_2$**

The sum of $CO_2$.anthr and $CO_2$.nee concentrations provides the regional contribution to the $CO_2$ concentration at JFJ (i.e., $CO_2$.regional). Together with the simulation-specific background for either FLEXPART-COSMO or STILT-ECMWF this yields the total $CO_2$ concentration (i.e., $CO_2$.total) at JFJ.

**2.4 Model-based $\delta^{13}$C-$CO_2$ estimation**

The stable carbon isotope ratio of $CO_2$ is referred to as $\delta^{13}$C-$CO_2$, or $\delta^{13}$C in short. The estimation of the mixed $\delta^{13}$C-$CO_2$ source signature ($\delta^{13}C_m$) and ambient $\delta^{13}$C-$CO_2$ isotope ratios ($\delta^{13}C_a$) is based on the $CO_2$ concentration simulations. All $\delta^{13}$C-$CO_2$ estimates are given in permille (‰) relative to the Vienna Pee Dee Belemnite (VPDB) reference standard. Further information on stable isotope expressions and definitions are available in Coplen (2011).

**2.4.1 Mixed source signature ($\delta^{13}C_m$)**

The absolute values of simulated $CO_2$ concentrations per source and sink category $i$, $|f_{s,i}|$, were weighted with category-specific source signatures, $\delta^{13}C_{s,i}$, to retrieve a mixed source signature, $\delta^{13}C_m$ according to Eq. (1) using the $\delta^{13}C_s$ literature-based assumptions summarized in Table 2 and Table 3. The simulated anthropogenic $CO_2$ data were disaggregated based on fuel type (Table 2) rather than sectorial processes, because $\delta^{13}C_s$ can best be attributed as a function of fuel type. For ecosystem fluxes, a seasonal cycle in $\delta^{13}C_s$ was assumed (Table 3). Following the

reasoning of Vardag et al. (2016), the $CO_2$.gee was treated as source, i.e., its absolute value, was considered, along with the $\delta^{13}C_s$, using a reversed sign in Eq. (1).


**Table 2.** Fuel type-specific $\delta^{13}C_s$ assigned to the simulated anthropogenic $CO_2$ categories.

| $CO_2$.anthr | $\delta^{13}C_s$, ‰ |
|---|---|
| **$CO_2$.fuel** | |
| gas, natural | −44.0 |
| gas, derived | −44.0 |
| coal, hard | −24.1 |
| coal, brown | −24.1 |
| coal, peat | −24.1 |
| oil, heavy | −26.5 |
| oil, light | −26.5 |
| oil, mixed | −26.5 |
| bio, gas | −60.0 |
| bio, solid | −24.1 |
| bio, liquid | −26.5 |
| **$CO_2$.cement** | |
| cement | −0 |

Assumptions for fossil and cement sources are based on Andres et al. (1994). Gaseous fuels are characterised by a large range (−15 to −85 ‰) as reviewed by Sherwood et al. (2017), with a mean of −44 ‰. The biogas signature is based on measurements of $\delta^{13}$C-CH$_4$ released by cows, a biogas production plant, and waste-water treatment (Hoheisel et al., 2019; Levin et al., 1993). The values are in line with microbial $\delta^{13}$C-CH$_4$ reviewed by Sherwood et al. (2017). $CO_2$.cement includes industrial emissions from cement production (NMM) alongside two minor contributors (CHE, IRO), as detailed in Table S1.

**Table 3.** Assumptions for ecosystem $\delta^{13}C_s$, based on Ballantyne et al. (2010 and 2011) and Vardag et al. (2016).

| Months | $\delta^{13}C_s$, ‰<br>$CO_2$.resp | $\delta^{13}C_s$, ‰<br>$CO_2$.gee |
|---|---|---|
| January | −27 | −25 |
| February | −26 | −24 |
| March | −25 | −23 |
| April | −24 | −22 |
| May | −23 | −21 |
| June | −22 | −20 |
| July | −22 | −20 |
| August | −23 | −21 |
| September | −24 | −22 |
| October | −25 | −23 |
| November | −26 | −24 |
| December | −27 | −25 |

$$\delta^{13}C_m = \frac{\sum_{n=1}^{i} (|f_{s,i}| \times \delta^{13}C_{s,i})}{\sum_{n=1}^{i} (|f_{s,i}|)} \qquad (1)$$

### 2.4.2 $\delta^{13}$C-$CO_2$ background estimate ($\delta^{13}C_b$)

The Jena CarboScope (JCS) $CO_2$ background concentration simulation for JFJ serves as $f_b$. The $\delta^{13}$C-$CO_2$ background value, $\delta^{13}C_b$, is estimated thereof through a relationship between observations of $CO_2$ and $\delta^{13}$C-$CO_2$ in background air, derived - similar to Vardag et al. (2016) - by applying yearly linear regression fits between measurements of $CO_2$ concentration and $\delta^{13}$C-$CO_2$ under free troposphere conditions at JFJ. The regression fits and background are provided in Figure S4 (exhibiting a seasonally varying background value).





### 2.4.3 Atmospheric $\delta^{13}C$-$CO_2$ estimates ($\delta^{13}C_a$)

The mixed source signature, $\delta^{13}C_m$, derived in Eq. (1) was combined with the background estimates ($f_b$, $\delta^{13}C_b$) in
order to derive estimates of atmospheric $\delta^{13}C$-$CO_2$ isotope ratios at JFJ, $\delta^{13}C_a$, following Eq. (2). Note that, contrary
to Eq. (1), $CO_2$.gee is considered as effective sink in Eq. (2), which is further detailed in Vardag et al. (2016).

$$\delta^{13}C_a = \frac{(f_b \times \delta^{13}C_b) + (\sum_{n=1}^{i}(f_{s,i}) \times \delta^{13}C_m)}{f_b + \sum_{n=1}^{i}(f_{s,i})} \qquad (2)$$

## 2.5 Observation-based $\delta^{13}C$-$CO_2$ estimation

Observation-based mixed source signature, $\delta^{13}C_m$, were derived using a moving Keeling-plot approach following
the example of Vardag et al. (2016) and using JFJ specific fitting and filtering criteria, as detailed in section 3.2.4.

## 2.6 Observations

The $CO_2$ concentrations and $\delta^{13}C$-$CO_2$ isotope ratios were continuously measured at JFJ by quantum cascade laser
absorption spectroscopy (QCLAS) during the period 2009–2017. The custom-built QCLAS instrument (Nelson et
al., 2008; Tuzson et al., 2008b, 2008a, 2011; Sturm et al., 2013) provides high-precision data for the three main
$CO_2$ isotopologues ($^{12}C^{16}O_2$, $^{13}C^{16}O_2$ and $^{12}C^{16}O^{18}O$), and therefore, it allows simultaneous determination of the $CO_2$
concentration and the $\delta^{13}C$-$CO_2$ and $\delta^{18}O$-$CO_2$ ratios at 1 s time resolution. The $CO_2$ dry air mole fractions (µmol
mol$^{-1}$) are reported in units of parts per million (ppm) on the World Meteorological Organization (WMO) $CO_2$
X2007 scale, while the isotope ratio values are given in ‰, relative to the Vienna Pee Dee Belemnite (VPDB)
reference standard. The instrument was configured as described in Tuzson et al. (2011) during 2009–2011.
Hardware and calibration strategy were revised during an upgrade in 2012, as described in Sturm et al. (2013) to
improve long-term precision, stability, and SI-traceability. Furthermore, the instrument participated in the
WMO/IAEA Round Robin 6 Comparison Experiment to assess the instrument capability to maintain the link to the
WMO recommended level under field operation (NOAA, 2015). Stable operating conditions guarantee a precision
of 0.02 ‰ for $\delta^{13}C$-$CO_2$ and 0.01 ppm for $CO_2$ at an optimum averaging time of 10 min. However, laboratory
temperature instabilities during 2016–2017 adversely affected instrument performance. $CO_2$ concentrations were in
addition determined at 1 min time resolution by a commercial cavity ring-down spectrometer (CRDS, G2401,
Picarro Inc., USA) since 2010, likewise linked to the WMO CO2 X2007 scale. These data are available as ICOS
product (Emmenegger et al., 2020). The mean difference (1σ) between the 10 min averaged CRDS and QCLAS
data is 0.1±0.4 ppm for the entire observation period. Besides the in-situ measurements, air samples were collected
in triplicate every second Friday at around 7 AM local time, i.e., at a time when the JFJ site predominantly
experiences lower free troposphere conditions (Herrmann et al., 2015). $CO_2$ concentration, $\delta^{13}C$-$CO_2$ and $\delta^{18}O$-$CO_2$
in the flask samples were analysed at Max Planck Institute for Biogeochemistry (MPI-BGC) in Jena as described
in Van Der Laan-Luijkx et al. (2013). A comparison with the QCLAS measurements for 2009–2017 indicates very
good agreement (no apparent systematic bias as function of time or signal intensity and overall agreement within
the extended compatibility parameters of the WMO (±0.2 ppm for $CO_2$, ±0.1 ‰ for $\delta^{13}C$-$CO_2$). The flask data,
which correspond to background conditions at JFJ as defined by the sampling time, are used to construct $\delta^{13}C_b$.





## 2.7 Time-series Analysis

Time series analysis was performed using R programming language, v3.6.1 (R Core Team, 2019), deploying
available R packages (https://cran.r-project.org) as well as custom developed scripts. While FLEXPART-COSMO
simulations provide 3-hourly averages, STILT-ECMWF provides instantaneous snapshots every $3^{rd}$ hours. STILT-
ECMWF simulations were interpolated between the 3-hourly nodes for comparison with 3-hourly averages of
observational data. For comparing the observations with the LPDM model output, we use 3-hourly and monthly
averages of the QCLAS measurements. Furthermore, a common JCS-based background is subtracted from the
measurements. The STILT-ECMWF JCS-based background is preferred as common background for this particular
assessment over the FLEXPART-COSMO background owing to the higher short-term variations in the latter
(compare Figure S3a). The background-subtracted data set is referred to as "regional observations".

## 3. Results and Discussion

### 3.1 Regional CO$_2$ simulations at JFJ

#### 3.1.1 Monthly time-scale

##### A) Planetary boundary layer influence at JFJ

Air mass transport dynamics determine the exposure of the receptor site JFJ to air masses from the planetary
boundary layer (PBL). Thus, together with the source or sink strength in the footprint region, they drive the regional
contributions to the CO$_2$ concentrations, and are discussed upfront. Previous analyses of tracers (e.g., radon and
CO/NO$_y$) by Herrmann et al. (2015) suggested that, compared to winter (December–February), the PBL-influence
at JFJ is enhanced by 1.5 to 2.5-fold in April and August/September, and by 3 to 4-fold from May–July. To isolate
the influence of seasonally varying transport, we performed dedicated simulations where CO$_2$ fluxes were assumed
constant in space and time (see Appendix A1). This analysis revealed a 2 to 3-fold larger simulated PBL-influence
in summer compared to winter for both models. Diurnal variations were most pronounced in summer, indicating a
1.4-fold larger PBL-influence during the afternoon and evening (maximum at ~16:00 h, UTC+1) compared to the
morning (minimum at ~10:00 h, UTC+1). A larger PBL-influence in May and September for STILT-ECMWF
compared to FLEXPART-COSMO appears to be a peculiarity of using ECMWF fields and may reflect the less-
well resolved transport in complex terrain in the coarser resolution data from ECMWF. Additional differences
appear related to the smaller domain size and shorter backward integration used for FLEXPART-COSMO, which
are directly associated with smaller integrated surface CO$_2$ fluxes.

##### B) Regional CO$_2$ concentration observations and simulations

Simulated CO$_2$.regional for 2009–2017 is compared with the respective regional CO$_2$ concentration observations in
Figure 1 (multi-annual monthly means of 3-hourly). The CO$_2$.regional observations show a minimum in June and a
maximum in October and November, both with an amplitude of about 1.8 ppm. Subpanels present the corresponding
simulated anthropogenic (CO$_2$.anthr) and ecosystem components (CO$_2$.nee, CO$_2$.gee, CO$_2$.resp).

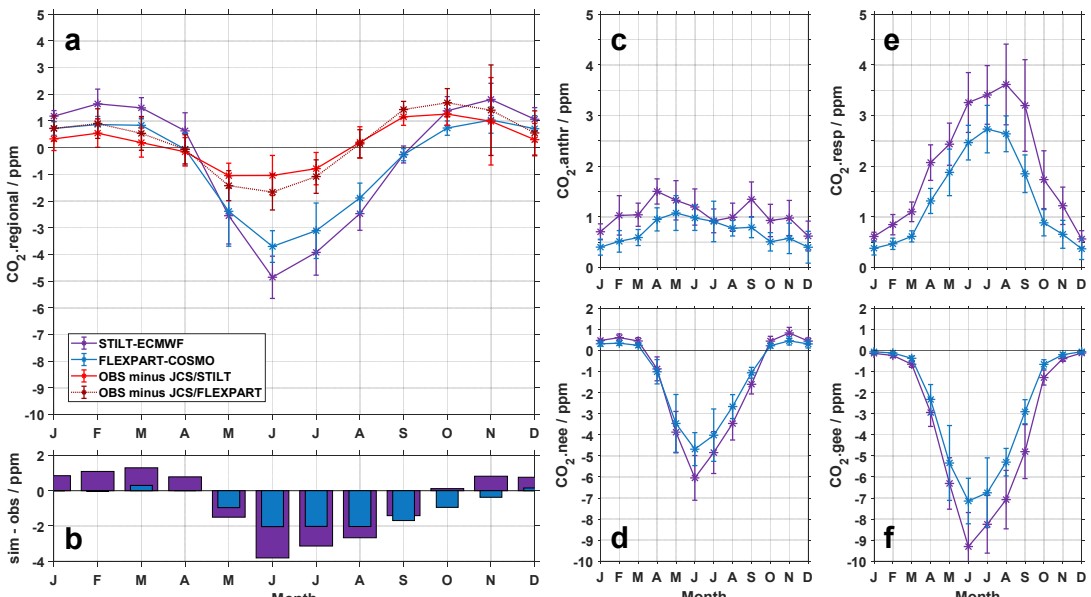

**Figure 1.** Multi-annual monthly means of 3-hourly regional $CO_2$ simulations compared to observations (2009–2017). $CO_2$.regional **(a)**, and its components $CO_2$.anthr **(c)**, and net ecosystem exchange ($CO_2$.nee) **(d)**. The difference between simulations (sim) and observations (obs) are presented in **b)**. $CO_2$.nee is composed of **e)** gross ecosystem respiration ($CO_2$.resp) and **f)** gross ecosystem exchange ($CO_2$.gee), i.e. gross uptake. Error bars represent 1SD of the multi annual means and reflect the year-to-year variability for 2009–2017.

While $CO_2$.anthr and $CO_2$.nee together constitute $CO_2$.regional, the sum of ecosystem components $CO_2$.gee and $CO_2$.resp results in $CO_2$.nee. The minimum in June as observed in the measurements is well represented by the models, though the amplitude is overestimated. The October/November maximum is delayed in both models by about one month. A local minimum in December/January is seen in observations and models. The winter minimum in the regional signal reflects the limited influence of PBL air masses at JFJ during this period of the year, and coincides with a minimum in $CO_2$.anthr (Figure 1c) and ecosystem $CO_2$ (Figure 1d-f). The models thus appear to represent the processes contributing to the seasonal variability of the regional $CO_2$ signal at JFJ quite realistically. Noteworthy, the seasonal trends of the regional signal, in particular the local winter minimum, differ from those in the large-scale $CO_2$ background concentrations, which show a minimum in August, two months later than the regional signal, and only one maximum in March/April, as shown by Sturm et al. (2013).

Regarding $CO_2$.anthr (Figure 1c), we conclude that the reduced transport of PBL air to JFJ during December/January outweighs a maximum in anthropogenic surface emission fluxes related to enhanced fuel use for heating during the cold season. Instead, $CO_2$.anthr simulations reach a maximum at JFJ in spring (April/May) in both models, resulting from still relatively large anthropogenic surface emissions and generally more unstable atmospheric conditions due to rising surface temperatures and sustained colder temperatures aloft. The STILT-ECMWF simulations comprise a second $CO_2$.anthr maximum in autumn (September), which is in line with the simulated PBL-influence (Appendix A1).

Given that ecosystem contributions quantitatively dominate the regional contributions to $CO_2$ concentrations during summer, we reiterate that the $CO_2$.nee simulations depend on the parameterization of



ecosystem respiration and uptake fluxes in VPRM. The parameterization accounts for environmental factors such
as temperature, radiation, and through MODIS derived enhanced vegetation index (EVI) and land surface water
index (LSWI) also for soil moisture (Mahadevan et al., 2008). Warmer temperatures generally lead to enhanced
gross ecosystem fluxes ($CO_2$.resp and $CO_2$.gee) in summer compared to winter. These trends are indeed reflected
in the simulations for JFJ (Figure 1d-f). The strong negative regional $CO_2$.nee from March to October is a result of
only partial compensation of uptake ($CO_2$.gee) by respiration ($CO_2$.resp). The $CO_2$.gee minimum in June does not
coincide with the $CO_2$.resp maximum in July/August. This may be explained by the fact that respiration is strongly
dependent on temperature, and July and August typically show the highest average temperatures in the relevant
footprint region. Ecosystem uptake, on the other hand, has a more complex relationship with temperature (drops off
when too hot), radiation (actually largest in June), water availability (usually decreasing during the summer), and
plant phenology (e.g., Bonan, 2015; Mahadevan et al., 2008).

The simulations qualitatively satisfy our expectations. However, the overestimation of the amplitude in
summer and early autumn by the two models merit further discussion of potential contributions to this mismatch,
which includes uncertainties in the transport model or in the spatio-temporal flux distribution. A quantitative
assessment is available in section 3.2.3.

1) *Transport Dynamics*: The fluxes computed by VPRM together with the air mass transport dynamics determine
the final seasonality of the ecosystem-related $CO_2$ contributions at JFJ.

    a. It has been reported by Denning et al. (1999) that the signal from respiration $CO_2$ is amplified over flat
terrain, because respiration dominates at night when the boundary layer is shallow. This observation is
referred to as "*rectifier effect*". At JFJ, we likely observe the inverse situation, a "*fair-weather effect*",
as warm and sunny afternoons favour PBL-influence at JFJ, while low irradiation periods (nighttime,
winter) limit the PBL-influence. Vertical atmospheric transport and photosynthetic activity (uptake) co-
vary and are both largest on sunny days. In contrary, ecosystem respiration is active independently of
light condition (day/night) and, to a smaller degree, during colder periods, when PBL-influence is
limited at JFJ. Such "*fair-weather effect*" may be inadequately captured in the models.

    b. The simulations for JFJ indicate that a considerable fraction of ecosystem $CO_2$ originated from fluxes
within the last few hours before arrival at JFJ and at distances shorter than 100 km from the site
(predominantly north of JFJ). We find that this "nearby" contribution is particularly pronounced in
summer, whereas cold season sampled air masses are rather associated with a much wider concentration
footprint and are less dominated by those "nearby" vegetation fluxes. In addition, the nearby vegetation
fluxes seem artificially enhanced by the limited spatial resolution of the vegetation maps (see also 2c).

2) *VPRM*: An overestimation of the $CO_2$.gee or an underestimation of $CO_2$.resp may be associated with harvesting
activities and drought stress, which are not well reflected in the current parameterisation of VPRM, as well as
the spatial representation of vegetation maps and temperature profiles.

    a. Harvesting usually results in a change of the Enhanced Vegetation Index (EVI) derived from the
MODIS observations. While the reduced ecosystem uptake due to harvesting is thus in principle already
represented in VPRM, the agricultural biomass left behind after the harvest may lead to increased



respiration. VPRM is unlikely to capture this latter process with its simple linear dependence of respiration on temperature.

  b. Water stress (drought), can lead to altered respiration and uptake fluxes (e.g., Ramonet et al. (2020) or Gharun et al. (2020)), but is not explicitly included in VPRM.

405   c. Owing to the smoothed topography and vegetation maps in the models, the effective temperatures in alpine vegetation is likely not well represented and, moreover, the temperature-parameterisations in VPRM is not be optimized for alpine vegetation. No systematic bias net ecosystem exchange is apparent for ecosystem simulations with STILT-ECMWF for other observational sites in Europe (data available at the ICOS Carbon Portal), suggesting that the discrepancy is predominantly linked to JFJ's location 410 in complex terrain. Indeed, summer discrepancies appear to be comparatively large at JFJ (3580 m a.s.l.) even when considering other mountain stations, such as Monte Cimone (~2000 m a.s.l., Italy) or Puy de Dôme (~1500 m a.s.l., France), which are characterized by lower altitude and less complex topography compared to JFJ.

 3) *EDGAR*: A mismatch between $CO_2$.regional simulations and observations may also result from biases in the 415  $CO_2$.anthr signal. However, as quantified in see subsection 3.2.3, an increase of $CO_2$.anthr by a factor of 3 to 4 would be required in order to compensate the summer mismatch. Further, the discrepancy during summer is much larger than that during winter when $CO_2$.anthr contributes the largest share, and we consider is thus unlikely that $CO_2$.anthr is the main driver of the summer mismatch. As JFJ is also a popular destination for touristic daytrips, local emissions from tourists and the JFJ infrastructure itself cannot be excluded. A recent 420 study by Affolter et al. (2021), however, showed that this effect is expected to be well below the discrepancy between observations and simulations found here.

### C) Composition of simulated anthropogenic and ecosystem $CO_2$

Ecosystem contributions to $CO_2$ concentrations outweigh the anthropogenic ones at JFJ most of the year if we consider the multi-annual monthly means (Figure 1). For instance, gross respiration contributions to $CO_2$ 425 concentrations are at their maximum 3-4 fold the anthropogenic ones during summer. However, gross respiration is overcompensated by an up to two fold gross uptake in summer. During the colder period, gross respiration dominates the net ecosystem exchange and equals roughly the amounts of anthropogenic $CO_2$. While on a global scale monthly ecosystem fluxes indeed outweigh anthropogenic $CO_2$, this is not the case for urban areas. For instance, Vardag et al. (2016) suggests that on cold winter days, the $CO_2$ share in an urban environment in Germany 430 (Heidelberg) is 90–95 % fuel-related, which is ~2-fold the $CO_2$.anthr fraction compared to JFJ). Nevertheless, also in Heidelberg ecosystem contributions can make up 80 % in summer, similar to our simulations for JFJ.

  In Figure 2a/b we present the ecosystem contributions at JFJ split for the considered vegetation types (multi-annual monthly means for 2009–2017, available for STILT-ECMWF only). For summer, the largest fractions of simulated $CO_2$.resp are related to cropland (~50 %), followed by forest (~30 %) and grassland (~10 %). During 435 winter, the cropland share increases, while the mixed forest share decreases. This may be a result of the above discussed change of footprint area from regional (cropland) in winter to more local (mixed forests) in summer. For $CO_2$.gee, it is important to consider that absolute quantities approach zero during the cold season, and relative fractions are most meaningful in summer. The $CO_2$.gee generally displays a larger forest share in comparison to the


one of $CO_2$.resp, possibly as air masses travel through forest-rich vegetated areas during the last few hours before

440    reaching JFJ (which corresponds to daytime, when uptake is active). Furthermore, we see a shift in the relative

$CO_2$.gee share from cropland to forest from April to September, which is likely the result of vegetation dynamics,

considering that crops mature earlier in the year, and forests absorb carbon much longer during the growing season.

In Figure 2c/d we present the relative fractions of $CO_2$.anthr. The contributions associated with fossil

sources sum up to 90 % of $CO_2$.anthr. $CO_2$.anthr is dominated by $CO_2$ from liquid fuel use, in particular light and

445    heavy oil used for on- and off-road transport as well as domestic heating (~50 %). Further 25 % of $CO_2$.anthr are

related to natural gas, and only 10 % are attributed to solid fossil fuels, including a larger fraction of hard coal and

a smaller fraction of brown coal. Solid biomass, such as residential wood burning for domestic heating, contributes

10 % to $CO_2$.anthr. Non-combustion $CO_2$ from cement and other industry production amounts to 5 % of $CO_2$.anthr

at JFJ. Seasonal shifts are observed in the contribution of solid biomass (higher in winter, lower in summer) as well

as in relative fractions of light oil (higher in summer) and natural gas (lower in summer). The relative contributions

of FLEXPART-COSMO (not shown here) are very similar to the ones of STILT-ECMWF despite the differences

in the absolute quantities of $CO_2$.anthr between the two models (Figure 1), which, as discussed above, are primarily

driven by the model's implementation of transport dynamics.

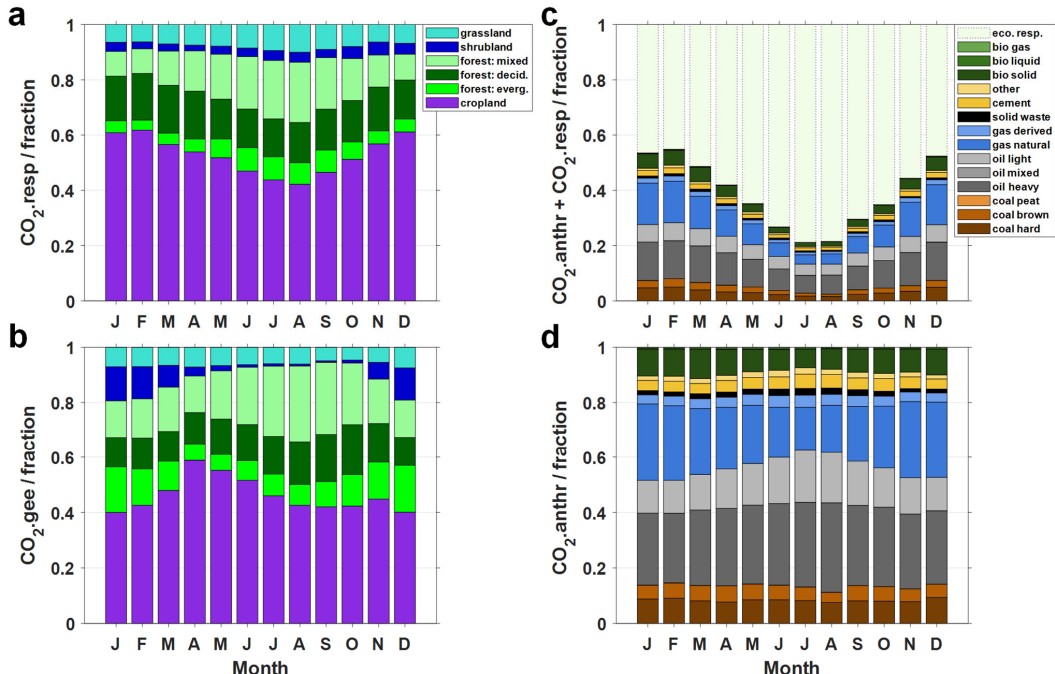

**Figure 2.** Simulated regional contributions to the $CO_2$ concentrations at JFJ (multi-annual monthly means of 3-hourly simulations, 2009–2017, STILT-ECMWF). a) gross ecosystem respiration per vegetation type ($CO_2$.resp), b) gross ecosystem exchange (uptake) per vegetation type ($CO_2$.gee), c) $CO_2$.anthr and $CO_2$.resp, d) $CO_2$.anthr per fuel-type. Maps of anthropogenic fluxes and vegetation distribution are provided in Figure S1 and S2.



### 3.1.2 Regression analysis of hourly-scale $CO_2$ simulations vs. observations

The model performance was further evaluated by comparing the 3-hourly simulated $CO_2$ concentration time-series with observations. In Figure 3 we present $CO_2$.total, which includes background ($f_b$) and regional contributions ($CO_2$.regional, i.e., the sum of $f_{s,i}$). In order to derive $CO_2$.total, the simulation-specific background (i.e., either
FLEXPART-COSMO or STILT-ECMWF) was added to the respective $CO_2$.regional data. Overall, the simulations capture the intensity and timing of individual regional short-term events at the models' 3-hourly time-resolution to a high degree, in addition to the good representation of annual and seasonal trends.

We assess the performance separately for the four seasons winter (December–February, or DJF), spring (March–May, or MAM), summer (June–August, or JJA) and autumn (September–November, or SON) for the
$CO_2$.regional signal, as summarized in Figure 4, and show a four-year subset for 2012–2015 in addition to the full nine-year observation period (2009–2017). The subset is of interest as it comprises a higher frequency and intensity of regional $CO_2$ at JFJ, in particular considering the winter of 2012/2013, and aside, measurements by QCLAS had the best performance during 2012–2015. We consider primarily the coefficient of determination, $r^2$, regression slope, and bias-corrected root mean square error (BRMS) in the assessment of the short-term variability.

The mean bias, labelled Y-X) and provided in Figure 5, is usually smaller than 1 ppm with the exception of summer, when the models exhibit a negative bias of up to 2.5 ppm. Removing this bias before calculating the root mean square error (RMSE) focusses onto the short-term variability. The BRMS ranges from 1.8 to 3.1 ppm $CO_2$, with lowest errors observed during winter and autumn, and highest errors in summer. For the 3-hourly data, both models reproduce the regional signal with similar quality. The $r^2$ is 0.44 for FLEXPART-COSMO and 0.41 for
STILT-ECMWF, meaning that the models explain about 40 % of the observed regional $CO_2$ variability at JFJ. Considering the complex topography and small amplitude of the regional signal, this is a very satisfactory result, and in line with comparable simulations by Henne et al. (2016), which were able to explain a similar fraction of variability in regional $CH_4$ at JFJ for the year 2012, after simulations optimization with respect to $CH_4$ emissions.

When analysing individual seasons, we find that the summer period is characterised by significantly lower
$r^2$ for the 3-hourly data compared to the other seasons, although, aside of above-mentioned negative bias, diurnal profiles in the observations during summer are well represented by the simulations. The slightly better performance for FLEXPART-COSMO compared to STILT-ECMWF in terms of mean bias and $r^2$ for 3-hourly data may be partly attributed to the higher spatial resolution that potentially allows for a better representation of thermally driven atmospheric transport in mountainous terrain during summer. Note that when adding model-specific JCS
background values to the regional simulations, $r^2$ values are substantially higher (~0.6–0.9, not shown), because a considerable part of variability in $CO_2$.total derives from seasonal variability and long-term trends.

The regression slopes represent the factors by which simulation and observation intensities agree with each other. For $CO_2$.regional, the intensity agreement (slope, ~0.9–1.5) varies as a function of season and model. Slopes are closest to 1 in autumn/winter, and, as for other regression parameters, larger discrepancies occur in
spring/summer. The spring/summer discrepancies are driven by negative excursions from the baseline in analogy to the larger warm season mismatch (discussed in 3.1.1) and higher mean bias. Again, note that we find the slopes for $CO_2$.total to be closer to 1 (~0.9–1.3, not shown), than those for the $CO_2$.regional, confirming the solid assumptions for the background $CO_2$ concentrations.



Figure 3. Time series of $CO_2$.total simulations with **a/c)** FLEXPART-COSMO and **b/d)** STILT-ECMWF compared to hourly observations. **a/b)** 2009–2017 (tick marks indicate January of each year), **c/d)** 2013. (JCS-based background is detailed in Figure S3a.)






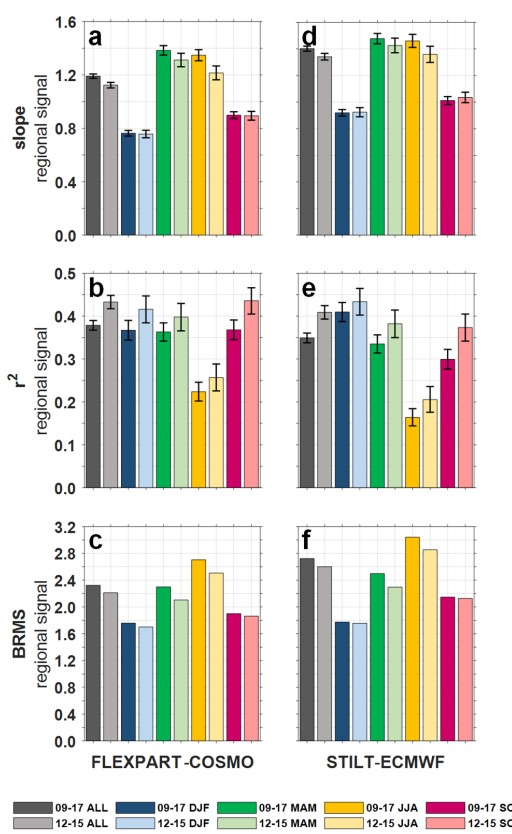

**Figure 4.** Summary of the regression analysis of $CO_2$.regional simulations vs. observation (data are based on 3-hourly time resolution; error bars = 95 % confidence interval). The parameters (slope, $r^2$ and bias corrected RMSE, i.e., BRMS) are presented for FLEXPART-COSMO (**a-c**) and STILT-ECMWF (**d-f**), including the full observation period, 2009–2017, and a 4-year subset (2012–2015).

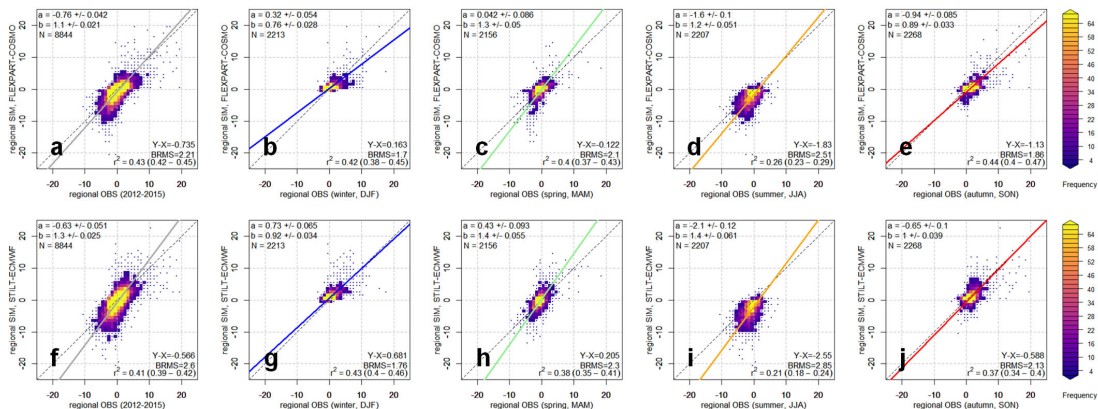

**Figure 5.** Heatmaps for $CO_2$.regional simulations (SIM) using FLEXPART-COSMO (**a-e**) and STILT-ECMWF (**f-j**), in comparison to regional components of observations (OBS) for 2012–2015, full year and per seasons, on 3-hourly time resolution. The STILT-ECMWF-based JCS background is subtracted from the observations to derive the regional component. The weighted least squares regression takes into account uncertainties in both data sets.



### 3.2 Atmospheric δ¹³C-CO₂

Although the challenges in terms of simulating regional signals at a high-alpine background site like JFJ are significant, JFJ is one of very few stations that offer continuous high frequency $\delta^{13}$C-CO$_2$ observations over multiple years. Thus, JFJ uniquely allows for evaluating model-based estimates of atmospheric $\delta^{13}$C-CO$_2$ and of mixed source signatures ($\delta^{13}$C$_m$) through comparison with atmospheric $\delta^{13}$C-CO$_2$ observations and thereof derived ("observation-based") $\delta^{13}$C$_m$ values using a moving Keeling approach.

**3.2.1 Regression analysis of hourly-scale atmospheric δ¹³C-CO₂ estimates vs. observations**

We evaluated the atmospheric $\delta^{13}$C-CO$_2$ isotope ratio estimates ($\delta^{13}$C$_a$), which are derived following Eq. (2) on a 3-hourly basis, through comparison with the QCLAS observations during the period 2012–2015 (Figure 6, Table 4).

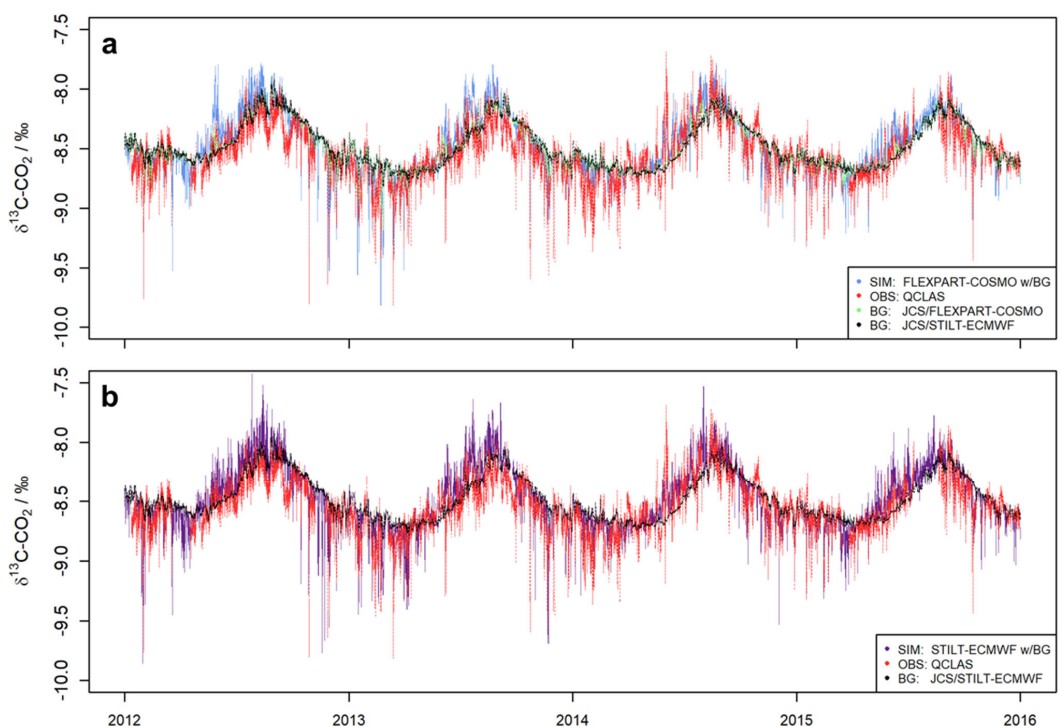

**Figure 6.** Time series of model-based and observed atmospheric $\delta^{13}$C-CO$_2$ for the years 2012–2015 (hourly observations). **a)** FLEXPART-COSMO, **b)** STILT-ECMWF; tick marks indicate January of each year. The background, $\delta^{13}$C$_b$, is presented in further detail in SI. Data are presented on hourly time resolution.

The simulated $\delta^{13}$C$_a$ time-series capture the observed variability in $\delta^{13}$C-CO$_2$ at JFJ well, in particular during the transition periods in spring and autumn. For most of the summer, however, the $\delta^{13}$C-CO$_2$ simulations are isotopically heavier than the observations, i.e. they appear more enriched in $^{13}$C. Despite an offset of ~0.15 ‰, the diurnal profiles in the observations during summer are well represented by the simulations, as also found for the CO$_2$ concentration. Generally, the discrepancy in $\delta^{13}$C appears to be larger for STILT-ECMWF compared to FLEXPART-COSMO, and thus the discrepancy in CO$_2$ concentrations itself likely contributes to the mismatch in $\delta^{13}$C-CO$_2$, as further assessed in 3.2.3, aside of uncertainties associated with assumptions for $\delta^{13}$C$_s$ and $\delta^{13}$C$_b$.





**Table 4.** Summary of statistics on atmospheric $\delta^{13}$C-$CO_2$ estimates and observations for the period 2012–2015. Values for min., max., median ($P_{50}$) and 25 and 75 percentiles ($P_{25}$ and $P_{75}$), mean (avg.) and 1SD are provided (hourly data). (see also Figure S6).

|  | min | $P_{25}$ | $P_{50}$ | $P_{75}$ | max | avg. | ±SD |
|---|---|---|---|---|---|---|---|
| **FLEXPART-COSMO** | −9.81 | −8.64 | −8.51 | −8.29 | −7.78 | −8.47 | ±0.24 |
| **STILT-ECMWF** | −9.86 | −8.65 | −8.52 | −8.29 | −7.42 | −8.47 | ±0.25 |
| **Observation (QCLAS)** | −9.81 | −8.64 | −8.47 | −8.29 | −7.78 | −8.47 | ±0.24 |

In addition to the total signal of atmospheric $\delta^{13}$C-$CO_2$ at JFJ we evaluate the regional contributions in Figure 7 and Figure 8. With regards to $\delta^{13}C_b$, a higher short-term variability was observed for FLEXPART-COSMO compared to STILT-ECMWF, as found in a similar manner for the JCS $CO_2$ background (Figure S3b), and the STILT-ECMWF-based background used for further calculations of regional components.

The regional estimates agree with the regional observations intensity within a factor of 0.7-1, depending on season. The BRMS is between 0.12 and 0.14 ‰. Similar to $CO_2$, for spring, autumn, and winter the models capture the short-term variability in $\delta^{13}$C-$CO_2$ better than in summer. Overall, the $r^2$ values are lower than for $CO_2$ (max. $r^2$ = 0.35 for FLEXPART-COSMO and 0.28 for STILT-ECMWF compared to about 0.4 for $CO_2$), which is not surprising given the uncertainties in the measurements as well as in the simulations, where, for instance, fixed source signatures were assumed. Despite the fact that model-based $\delta^{13}$C-$CO_2$ includes uncertainties of both, $CO_2$ simulation

(used to construct $\delta^{13}C_m$), $\delta^{13}C_s$ and $\delta^{13}C_b$, the relative performance decreased by only 20-30 %.

These results at JFJ were achieved with very low regional $CO_2$ signals, which, compared to the background ($\Delta CO_2$), reached at maximum 30 ppm. Instead, the previously conducted urban studies benefitted from much more pronounced $\Delta CO_2$ reaching up to ~150 ppm for both, Heidelberg (Vardag et al., 2016) and Downsview (Pugliese-Domenikos et al., 2019). However, they were limited regarding either the length of the observation period (few

months in Downsview), and/or the stringent data filtering (e.g., Vardag et al. (2016) discarded 85 % of the data and biased the urban data sets towards night-time observations, Pugliese-Domenikos et al., 2019 discarded 80% of the data for their isotopic mass balance approach). Contrary, the tall tower study in rural England was challenged by a low signal-to-background ratio ($\Delta CO_2$ reaching around 20 ppm), and isotope measurements were performed at low (weekly) time-resolution, although simulations are provided on hourly-scale (Wenger et al., 2019).

In comparison to the results from JFJ, Pugliese-Domenikos et al. (2019) reported an $r = 0.58$ ($r^2 = 0.3$), a root mean square error (RMSE) of 1.05 ‰ and a mean bias of 0.04 ‰ for a single month (January) for $\delta^{13}$C-$CO_2$. Wenger et al. (2019) do not provide any regression parameters for their model-observation comparisons; however, they observed large uncertainties in the $\delta^{13}$C-$CO_2$ estimation using a Monte Carlo approach. They related a part of their uncertainty for the $\delta^{13}$C-$CO_2$ estimates to the influence of ecosystem processes and the dominance of ecosystem

fluxes on the regional $CO_2$ observations and simulations at the rural tall tower site. Overall, the JFJ results are very well in line with previous findings despite the more remote location and correspondingly smaller magnitudes of regional signals at JFJ.





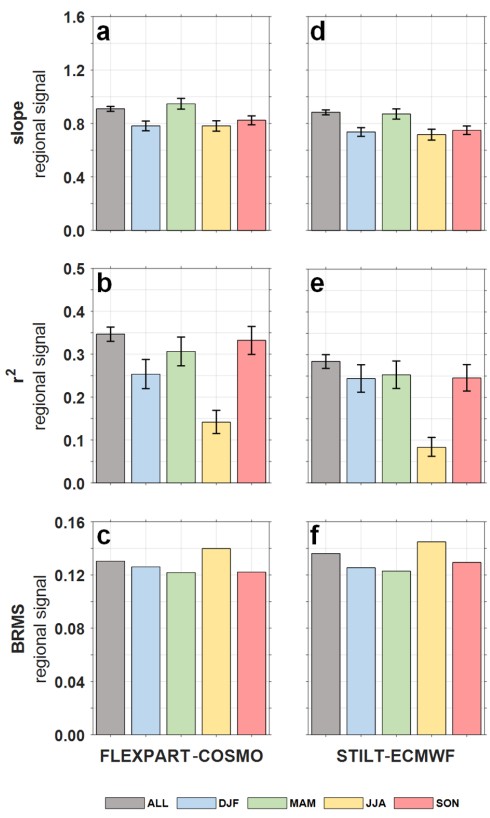

**Figure 7**. Summary of the regression analysis of $\delta^{13}$C-CO$_2$ estimation vs. observation (data are based on 3-hourly time resolution; error bars = 95 % confidence interval). Performance parameters (slope, r$^2$ and bias corrected RMSE (i.e., BRMS)) are presented for the 4-year subset of the observation period (2012–2015) for FLEXPART-COSMO (**a-c**) and STILT-ECMWF (**d-f**), across all year ("ALL"), and per season (DJF, MAM, JJA, SON).

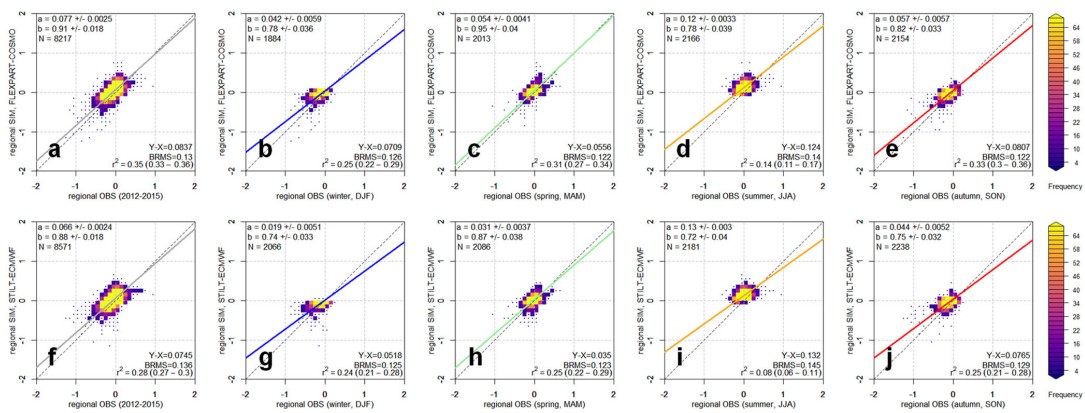

**Figure 8.** Heatmaps of model-based regional $\delta^{13}$C-CO$_2$ (SIM) vs. observation (OBS) (3-hourly data), for FLEXPART-COSMO (**a-e**) and STILT-ECMWF (**f-j**), during 2012-2015, for the full year (grey), and per season (DJF (blue), MAM (green), JJA (orange), SON (red)). Uncertainties in x- and y-axes are taken into account in the weighted least squares regression applied here.

### 3.2.3 Sensitivity of $\delta^{13}$C-CO$_2$ estimates to different model assumptions

*A) $\delta^{13}C_s$ assumptions*

The mixed source signature estimates ($\delta^{13}C_m$) as derived in Eq. (1) are presented in Figure 9 on a 3-hourly timescale (monthly data are provided in Figure S5). The estimated average $\delta^{13}C_m$ is around –24 ‰ and varies seasonally between around –22 ‰ in summer and –28 ‰ in winter, for both, FLEXPART-COSMO and STILT-ECMWF. Extreme values during particular events on 3-hourly time resolution reach –35 ‰ when they are heavily impacted

by anthropogenic fuel emissions including a larger fraction of natural gas (~50% of regional CO$_2$), and values between –17 to –12 ‰ when impacted by cement production (~30%). The $\delta^{13}C_s$ from cement production originates from carbonates, which are characterised by a similar isotope composition as the carbonaceous VPDB reference material itself. Consequently, the $\delta^{13}C_s$ for cement-related CO$_2$ is 0 ‰. Although cement-related CO$_2$ contributions to CO$_2$.regional at JFJ are about one order of magnitude smaller than from fuel burning or ecosystem processes, the

influence of cement on $\delta^{13}C_m$ is clearly visible in the model-based data in Figure 9. These cement-related peaks in $\delta^{13}C_m$ are, however, absent in $\delta^{13}C_a$ (Figure 6), simply because even the most intense cement signals at around 1-2 ppm are much smaller than other CO$_2$ contributions. Thus, when mixed with the background, the signal is diluted.

The $\delta^{13}C_s$ values, which are underlying the $\delta^{13}C_m$, represent the best available information in the scientific literature. However, while we use static assumptions, these values may vary in reality with air mass source region

(footprint) and over time. Further uncertainties may arise from assumed ecosystem $\delta^{13}C_s$. For instance, C4 plants are not explicitly represented in our model as a dedicated vegetation type with known spatial distribution. Yet their contribution to average ecosystem $\delta^{13}C_s$ is captured in the data of Ballantyne et al. (2010 and 2011), which are underlying the assumptions in Table 3, as these are derived from ambient measurements in mixed C3/C4 ecosystems representative for the Northern Hemisphere. In the footprint region of JFJ, C4 plants are mainly present in cropland

due to maize production. For the year 2017, EUROSTAT reports that the grain maize production made up around 21 % of the overall grain and cereal production by weight, within EU-28. Of all cropland, roughly 35 % on a land surface basis is assigned to grain and cereals. Applying a simple "back-of-the-envelope" calculation, this equates to ~7 % C4-related CO$_2$ fluxes within the European Union, as a yearly average. Because maize production is primarily relevant during the spring and summer, the fraction would be enhanced for this period of the year. Replacing 7 %

of the C3-related CO$_2$ with C4-related CO$_2$ would marginally change the source signature of crops (< 1 ‰, and that of the overall ecosystem signal by even less); however, generally $\delta^{13}C_m$ would become more enriched and thus the discrepancy between model and observations larger. Reducing a potential C4-related CO$_2$ fraction instead would make $\delta^{13}C_m$ less enriched and thus bring the simulations data into slightly better agreement with observations at JFJ. Indeed, the ecosystem assumptions for the Northern Hemisphere are based on data collected in the USA and might

be characterised by a higher C4 fraction than the footprint region for JFJ.

Vardag et al. (2016) report a measurement-based mean source signature ($\delta^{13}C_m$) of −26 ‰ in summer and about −32 ‰ in winter for Heidelberg, which is isotopically lighter when compared to the simulated $\delta^{13}C_m$ for JFJ (−22 ‰ in summer, −28 ‰ in winter). The winter differences between Heidelberg and JFJ is reasonable as it may derive from larger ecosystem contributions at JFJ (50 %) compared to Heidelberg (5 %). The summer differences,

however, may, aside from summer overestimations of CO$_2$.regional at JFJ, result from uncertainties in the assumption for the ecosystem $\delta^{13}C_s$ including the uncertainty of the C4-related CO$_2$ fraction. Indeed also Vardag et

al. (2016) suggest that the assumption of $\delta^{13}C_s = -23$ ‰ for ecosystem $CO_2$ by Ballantyne et al. (2011) is too enriched for August and September in Heidelberg, and a more depleted assumption (through adjusting the seasonality in $\delta^{13}C_s$) would indeed also result in further agreement between model-based $\delta^{13}C$-$CO_2$ and observations

at JFJ.

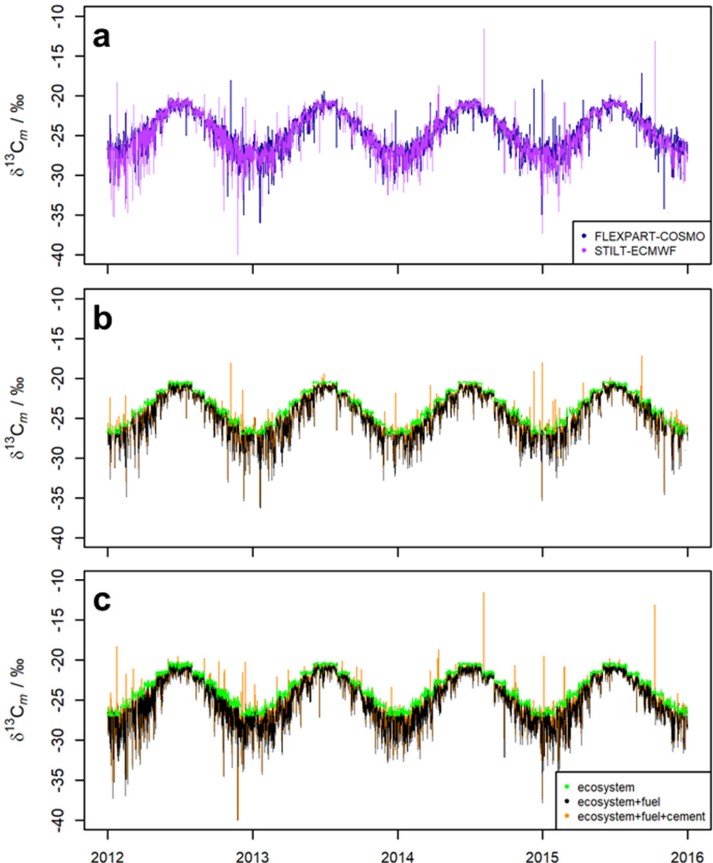

**Figure 9.** Time series **a)** model-based $\delta^{13}C_m$ (Eq. (1)), **b-c)** model-based $\delta^{13}C_m$ for different lumps of ecosystem-, fuel- and cement-related $CO_2$: **b)** FLEXPART-COSMO, **c)** STILT-ECMWF; hourly data are used; tick marks

indicate January of each year. (see also Figure S5)

### B) $\delta^{13}C_b$ assumptions

The simulated background ($\delta^{13}C_b$ in Figure 6 and Figure S3b), as estimated by the baseline $CO_2$ taken from the JCS assimilation system and the empirical $\delta^{13}C/CO_2$ relationship, tracks the evolution of the observed $\delta^{13}C$-$CO_2$ values

outside of the peaks very closely. Yet, slight inconsistencies are apparent from the use of yearly fits (Figure S3b and Figure S4). It appears that a more depleted $\delta^{13}C_b$ assumption during the second half of the year, for instance, by –0.15‰ during late summer (August) and early autumn (September), and a more enriched $\delta^{13}C_b$ assumption during the first half of the year, for instance by +0.05 to +0.10‰ in from January to March, ,would reduce the discrepancies between observations and simulations.





**_C) Sensitivity to CO$_2$ concentrations_**

Based on the discussion in section 3.1.1 we defined five scenarios, which aim to bring the simulated summer-time CO$_2$.regional concentrations into better agreement with the observations. In each scenario, we adjust one or a combination of CO$_2$ sources/sinks by a single scaling factor for the whole summer period (JJA) for the years 2012–2015, thereby removing the model bias.


- Scenario 1 (sc1): through increasing CO$_2$.anthr we simulate a bias in the anthropogenic emission fluxes or a wrong seasonal factor for CO$_2$.anthr during summer.
- Scenario 2 (sc2): through reducing both CO$_2$.resp and CO$_2$.gee we attempt to represent a general VPRM parameterisation or vegetation map representation issue.
- Scenario 3 (sc3): through reducing CO$_2$.gee we consider its potential overestimation by general VPRM parameterisation or vegetation map representation issue in analogy to sc2; specific only to CO$_2$.gee.
- Scenario 4 (sc4): through increasing CO$_2$.resp we consider its potential overestimation by general VPRM parameterisation or vegetation map representation issue in analogy to sc2; specific only to CO$_2$.resp.
- Scenario 5 (sc5): through modifying all signals at equal amounts (CO$_2$.anthr, CO$_2$.resp, CO$_2$.gee) we attempt
to represent a pure transport issue (i.e., overrepresentation of PBL-influence).

Scaling factors for each scenarios were derived by weighted least squares regression and presented in Table 5. The largest scaling factors of ~3-4 are found for CO$_2$.anthr, followed by CO$_2$.resp (~2), indicating that CO$_2$.anthr or CO$_2$.resp would need to be substantially increased in order to reduce the bias between model and observations.
Instead, a reduction (scaling factor ~0.7-0.8) would be required if only CO$_2$.gee was considered, and likewise a reduction in both, CO$_2$.resp and CO$_2$.gee (scaling factor ~0.7-0.8) in order to achieve a reduced CO$_2$.nee would lead to a reduced bias between model and observations.

**Table 5.** Scaling factors based on the weighted least squares regression fitting slope $b$, and intercept $a$ (in parenthesis), used to minimize the CO$_2$ model bias for JJA, 2012–2015.

| | FLEXPART-COSMO | STILT-ECMWF | CO$_2$ component |
|---|---|---|---|
| **base** | -- | -- | |
| **sc1 (anthr)** | 3.14 (a = 0.02) | 3.73 (a = -0.11) | × **CO$_2$.anthr** |
| **sc2 (nee)** | 0.80 (a = 1.04) | 0.72 (a = 1.22) | × **CO$_2$.resp** × **CO$_2$.gee** |
| **sc3 (gee)** | 0.79 (a = 0.45) | 0.74 (a = 0.49) | × **CO$_2$.gee** |
| **sc4 (resp)** | 2.08 (a = -0.88) | 1.98 (a = -0.56) | × **CO$_2$.resp** |
| **sc5 (trans)** | 0.82 (a = 1.29) | 0.74 (a = 1.54) | × **CO$_2$.anthr** × **CO$_2$.resp** × **CO$_2$.gee** |





We further evaluate the effect of these $CO_2$ adjustments on the estimated regional $\delta^{13}C$-$CO_2$ at JFJ in comparison to the observations. A representative set of results of the regression analysis is summarized in Table S4. Overall, we find that modifications in scenario 1 ($CO_2$.anthr) do not lead to an improvement in the agreement

between regional $\delta^{13}C$-$CO_2$ observations and simulations on 3-hourly resolution. Scenario 5 (transport) results only in small improvements with regards to the BRMS. While the other scenarios neither result in major adjustments, for scenario 3 ($CO_2$.gee) and scenario 4 ($CO_2$.resp) we observe small model improvements with slightly increased $r^2$, slightly reduced BRMS and a smaller bias (Y-X). Note, that the remaining bias depends on the fitting intercept assumptions of the scaling factor. Overall, these results indicate that the $\delta^{13}C$ agreement can be influenced through

modification of $CO_2$ contributions and discrepancies between observed and simulated $\delta^{13}C$-$CO_2$ are thus not only a direct result of uncertainties in source signature ($\delta^{13}C_s$) or background ($\delta^{13}C_b$) assumptions.

### 3.2.4 Observation-based source signature estimates

The model-based $\delta^{13}C_m$ may be compared to observation-based values, which are accessible through a "Keeling"- or "Miller-Tans" plot approach, however, only after strict pre-selection of conditions under which the underlying hypotheses are fulfilled. Detailed descriptions of pre-requisites and limitations of these models are available

elsewhere (Keeling, 1958; Keeling, 1961; Miller and Tans, 2003; Pataki et al., 2003; Zobitz et al., 2006; Ballantyne et al., 2011; Vardag et al., 2016), thus we provide only a brief discussion. Previous $\delta^{13}C_s$ studies have been successful in deriving observation-based $\delta^{13}C_m$ primarily under the following conditions: First, when measurements were taken rather close to a well-defined source location and using instrumentation with high precision (e.g., Pugliese et al.,

2017). Second, when a pronounced regional signal (referred to as $\Delta CO_2$ and computed as the difference between the $CO_2$ concentration at the site and background) with stable source composition was observed during stable background conditions and the regional ecosystem contribution to the observed $\Delta CO_2$ was comparatively low (e.g., Vardag et al., 2016). Such constrains substantially limit the number of regional events that can be effectively characterised at a given location. Intensities below $\Delta CO_2 = 5$ ppm, even at high precisions of 0.03 ‰ for $\delta^{13}C$-$CO_2$

and low $CO_2$ errors of 0.1 ppm, lead to significant fitting errors as assessed by Zobitz et al. (2006). Intensity-based filtering criteria have, therefore, been applied in previous studies (e.g. $\Delta CO_2 \geq 5$ ppm by Vardag et al. (2016), $\Delta CO_2 \geq 20$ ppm by Smale et al. (2019), $\Delta CO_2 \geq 30$ ppm by Pugliese-Domenikos et al. (2019), or $\Delta CO_2 \geq 75$ ppm by Pataki et al. (2003)), while at JFJ $\Delta CO_2$ reaches 30 ppm only during the most intense events. Most studies also focus on periods when photosynthetic uptake does not disturb the analysis, consequently biasing the data set to night-time.

Given that a classical day-/night splitting to filter ecosystem uptake is not applicable at JFJ as the received air masses are composed of integrated fluxes over day and night owing to the remote, high altitude location, such observation-based approaches are expected to be valid mainly during the cold period. However, and as discussed above, the PBL-influence at JFJ is at a minimum during and the cold season. For instance, regional $CO_2$ intensities at JFJ are at the maximum 30 ppm above the background for the 10 min averaged QCLAS data, and on average occur with

an intensity of $\geq 5$ ppm on 35 days per year during the cold period (range: 20–50 times). This includes events reaching $\geq 10$ ppm on 10 days per year (range: 2–20) and events reaching $\geq 15$ ppm on only 1–6 days per year. Intensities and frequencies, however, are even lower, when hourly averaged data are considered. Hence, because of the combination of low $\Delta CO_2$ and low event frequency, make Keeling/Miller-Tans methods to derive observation-based $\delta^{13}C_m$ particularly challenging at JFJ.



The high-precision of the $\delta^{13}$C-CO$_2$ measurements as well as the high time-resolution available from the QCLAS instrument allow to compensate the low $\Delta$CO$_2$ and to limit fitting uncertainties to some extent. This allows us to performed a moving Keeling-plot in analogy to Vardag et al. (2016), using various fitting and filtering criteria. We used a 5 hour window to conduct the moving Keeling fit on hourly averaged $\delta^{13}$C-CO$_2$ observations. Only fits with five data points were considered (i.e., no data gaps were allowed). In addition, we tested splitting the data set

into warm (Apr-Sept) and cold season (Oct-Mar), and demanding a minimum change in $\Delta$CO$_2$ of 3 ppm within the 5 hour window (with and without requiring a monotonous increase in concentration with time, threshold: 0.1 ppm). Finally, we filtered the resulting observation-based intercept value ($\delta^{13}$C$_m$) by the fitting error (4, 3, 2 and 1 ‰).

        Figure 10a shows model-based estimates in comparison to observation-based estimates, firstly without considering any predefined change in $\Delta$CO$_2$ and without filtering by the intercept error (referred to as "all"), and,

secondly, results obtained under more stringent criteria (minimum $\Delta$CO$_2$ change within a 5 h window of 3 ppm, maximum intercept error of 2 ‰ or 1 ‰)). Keeling fit intercepts ($\delta^{13}$C$_m$) obtained without predefined criteria and without error-based filtering do not provide meaningful data, as $\delta^{13}$C$_m$ is physically meaningful only between 0 ‰, corresponding to pure cement production plumes, and, –44 ‰ corresponding to pure gaseous fuel burning plumes (in a peculiar event, gaseous fuel burning CO$_2$ may reach –85 ‰). Most values are expected between –12 and –35

‰ based on the model results. Indeed, using predefined fit criteria and error-based filtering allows to generate physically meaningful $\delta^{13}$C$_m$ from the observations at JFJ, in line with previous findings by Vardag et al. (2016) and Pugliese-Domenikos et al. (2019). Overall, the observation-based $\delta^{13}$C$_m$ derived with a more stringent fitting approach are in good agreement with the model-based data (Figure 10a-d, Table 6), despite the substantial decrease in number of data points. Because different combinations of predefined criteria (minimum $\Delta$CO$_2$ or season-based

restrictions) and filtering (based on the intercept error) may be used when deriving observation-based $\delta^{13}$C$_m$, we present three scenarios. Figure 10b highlights that only filtering by intercept errors of 4, 3, 2 and 1 ‰ is an insufficient measure. Instead, Figure 10c shows the combined effect of requiring a change in $\Delta$CO$_2$ > 3 ppm and filtering by intercept errors, and Figure 10d presents data only for the cold period (Oct-Mar), limiting the disturbance of photosynthetic uptake, in addition to requiring a monotonous increase in $\Delta$CO$_2$ within the 5 h window (i.e., the

most stringent criteria). In all tested cases, the observation-based estimates exhibited a larger variability compared to the model-based data. However, we may generally conclude that either more stringent intercept error thresholds (such as 1 ‰ for the settings in Figure 10b), or, alternatively, limiting photosynthetic uptake (through demanding monotonous increase, and/or filtering for cold season or night-time) in combination with less stringent intercept errors (e.g., 2-3 ‰ in Figure 10d) appear to yield equally good results at JFJ. The latter, however, discards more

data. The same conclusion holds true when using 10-min averages instead of hourly data.

        A further disaggregation using mass balance approaches and assumptions for the end-members in order to learn more about the CO$_2$ regional composition in comparison to the simulated CO$_2$ regional composition from the observation-based approach was not attempted here, given the small number of observation-based $\delta^{13}$C$_m$ data points available, but may be the focus in future studies. However, we expect that it will remain challenging to disentangle

fuel and ecosystem respiration signals from observation-based $\delta^{13}$C$_m$ alone, considering that the simulated regional CO$_2$ fractions at JFJ indicate approximately equal amounts even during the winter, and that solid and liquid fuel emissions $\delta^{13}$C$_s$ end-member assumptions overlap with C3 plant respiration signatures.

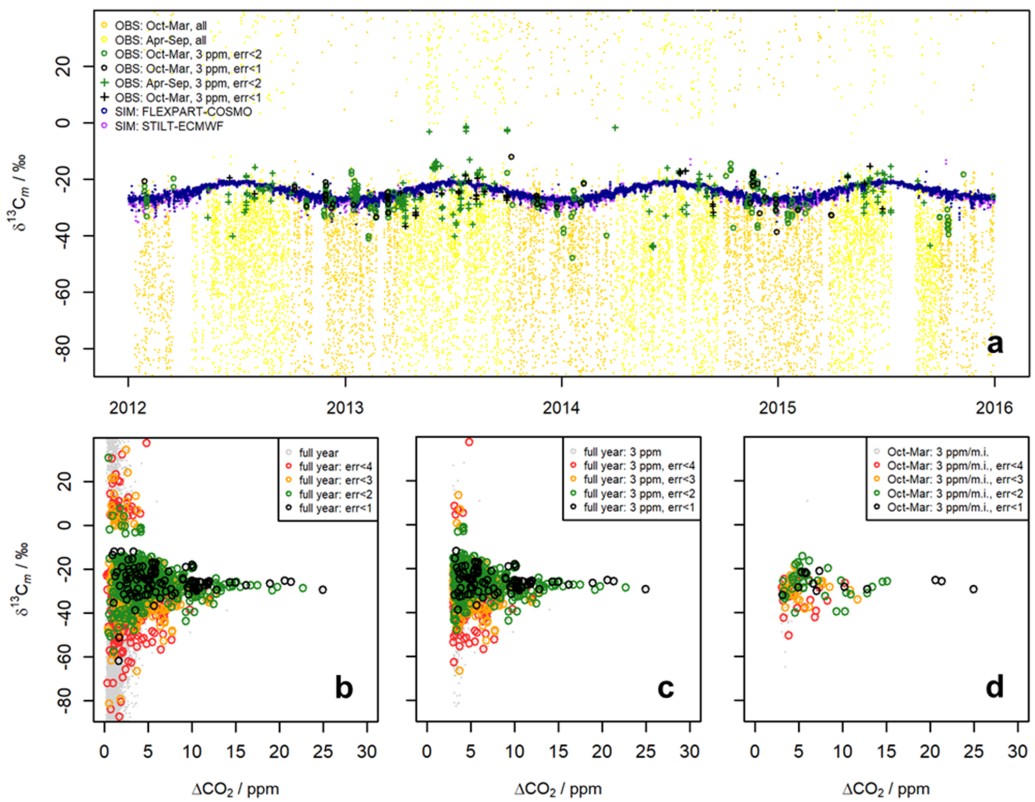

**Figure 10.** Observation-based mixed source signatures, $\delta^{13}C_m$, derived from a moving Keeling approach ("OBS") in comparison to model-based estimates ("SIM", FLEXPART-COSMO and STILT-ECMWF). **a)** time-series of $\delta^{13}C_m$ (tick marks indicate January of each year). "all" indicates that neither a minimum change in $\Delta CO_2$ was required, nor any filtering applied. Results when requiring a minimum change of 3 ppm in $\Delta CO_2$ within the 5 h window and a fit intercept error (err) < 2 ‰ and < 1 ‰ are provided as green and black markers (open circles represent Oct-Mar, crosses represent Apr-Sept). **b-d)** $\delta^{13}C_m$ hourly moving Keeling as a function of $\Delta CO_2$ for various criteria: **b)** filtering by intercept err < 4, 3, 2 and 1 ‰, **c)** demanding a minimum change in $CO_2$ of 3 ppm and filtering by intercept err < 4, 3, 2 and 1 ‰, **d)** demanding a monotonous increase in $\Delta CO_2$ of 3 ppm within the 5 h window and filtering by intercept err < 4, 3, 2 and 1 ‰.

**Table 6.** Summary statistics of $\delta^{13}C_m$ in ‰ (2012–2015).

|  | min | $P_{25}$ | $P_{50}$ | $P_{75}$ | max | avg. | ±SD |
|---|---|---|---|---|---|---|---|
| **FLEXPART-COSMO** | −35.95 | −26.38 | −24.26 | −22.08 | −17.16 | −24.29 | ±2.39 |
| **STILT-ECMWF** | −35.26 | −26.63 | −24.50 | −22.11 | −12.78 | −24.48 | ±2.57 |
| **OBS; 1‰ Figure 10b*** | −61.90 | −28.82 | −25.93 | −21.64 | −11.95 | −25.85 | ±6.85 |
| **OBS, 1‰ Figure 10c*** | −38.66 | −28.78 | −26.09 | −22.24 | −12.13 | −25.70 | ±4.88 |
| **OBS, 2‰ Figure 10d*** | −39.99 | −29.64 | −25.93 | −22.52 | −14.43 | −26.59 | ±5.56 |

*Figure 10b (err < 1‰, w/o $\Delta CO_2$ prerequisite, w/o seasonal filtering)

*Figure 10c (err < 1‰, $\Delta CO_2$ > 3 ppm, w/o seasonal filtering)

*Figure 10d (err < 2‰, $\Delta CO_2$ > 3 ppm (m.i.), Oct-Mar)



# 4. Conclusions

Greenhouse gas emissions source/sink identification and quantification at remote, high altitude sites is particularly challenging for broadly distributed, multi-source and multi-sink compounds such as $CO_2$. In addition, atmospheric transport simulations are highly challenged by complex topography. Despite these difficulties, the $CO_2$ simulations performed at 3-hourly basis for JFJ agree well with the observations during the multi-year period 2009–2017. Using Lagrangian particle dispersion models (LPDM), we were able to capture 40 % of the observed regional $CO_2$

variability. The results from the model configurations using two different LPDMs driven by output from two different numerical weather prediction systems, FLEXPART-COSMO and STILT-ECMWF, appear to differ primarily as a function of meteorological inputs and their spatial resolution (COSMO vs ECMWF), aside additional variations observed related to the domain size and backward integration time. The LPDM implementation (FLEXPART or STILT) itself contributes comparatively small differences.

Based on the regional $CO_2$ simulations, it appears that JFJ's high-altitude location predominantly experiences influences from the rather nearby (within 100 km) ecosystem. This is owing to the enhanced PBL-influence in summer, which overlaps with high ecosystem activity. Instead, the peak in anthropogenic fluxes during winter overlaps with substantially suppressed PBL-influence and a larger (regional) footprint. Therefore, through most of the year, the ecosystem $CO_2$ contributions composed mainly of cropland and mixed forest respiration and

uptake, outweigh the anthropogenic ones composed of 90 % fossil emissions and dominated by heavy and light oil, and natural gas. While the simulated composition resembles our hypothesis for JFJ, the extent to which ecosystem contributions outweigh anthropogenic ones is surprisingly large. Indeed, quantitatively, the models perform the $CO_2$ simulations best during winter and transition periods (spring/autumn). For the summer, the $CO_2$ simulations poorly reproduce the quantities despite the good qualitative agreement. The atmospheric transport models employed

apparently suffer from their relatively coarse spatial resolution, which deteriorates model performance in summer/fair-weather situations, when topography-induced convection is not captured very quantitatively during day-time. Increased model resolution and improved representation of the alpine boundary layer in both, the LPDMs and the driving numerical weather prediction models will be necessary to overcome this shortcoming and to allow for more quantitative conclusions when interpreting observations during the abovementioned conditions. However,

also the net ecosystem exchange fluxes themselves are a likely source of error through inaccurate spatial distribution and VPRM parameterisation of respiration and/or uptake fluxes for the (Alpine) vegetation following limited spatial resolutions of vegetation maps and possibly temperature profiles.

The simulations of regional $CO_2$ concentrations allow retrieving model-based mixed source signatures ($\delta^{13}C_m$) and atmospheric $\delta^{13}C$-$CO_2$ at JFJ. The latter agree remarkably well with the high frequency observations.

The overall $\delta^{13}C$-$CO_2$ correlation (28–35 %) remains only slightly lower than for $CO_2$ (41–44 %). In analogy to the findings for $CO_2$, also $\delta^{13}C$-$CO_2$ shows the lowest agreement between observations and simulations during the summer. We relate this primarily to the poorly reproduced $CO_2$ quantities in summer, although the assumption of source signatures ($\delta^{13}C_s$) as well as the estimate of the background ($\delta^{13}C_b$) provide additional uncertainties. For instance, our $\delta^{13}C_s$ estimates do not consider geographic variations in fuel specific $\delta^{13}C_s$ and ecosystem values are

not specific to photosynthetic pathways. Dedicated maps that allow to separate C3 and C4 vegetation in the VPRM



model would allow for even better representing the forward $\delta^{13}C_m$ of $CO_2$. Observation-based assessment of $\delta^{13}C_m$ are challenging at JFJ, owing to the low signal-to-background ratios and the integration of fluxes over day and night, which substantially limited the data set. Yet, the observation-based $\delta^{13}C_m$ agree well with the model-based estimates.

The simulated regional $CO_2$ composition at JFJ suggests that further analyses would benefit from a multi-tracer approach, in combination with the herein presented data. Additional parameters may include CO, atmospheric potential oxygen (APO), and $^{14}C$ as combustion or fossil fuel tracer; and carbonyl sulphide (COS) and $\delta^{18}O$-$CO_2$ as ecosystem tracers. Indeed, CO, APO, COS and $\delta^{18}O$-$CO_2$ observations are available at high time-resolution at JFJ and may be investigated in future, although determining their regional and background contributions will remain challenged by the low signal-to-background ratios. The bi-weekly integrated $^{14}CO_2$ data, currently available for JFJ,

instead do not allow distinguishing regional from background contributions. Highly time-resolved $^{14}CO_2$ measurements or grab sampling during periods with intense regional $CO_2$ influences would be highly valuable and is foreseen to be implemented at JFJ as part of the European-wide flask sampling strategy of the ICOS Research Infrastructure in future.





## Appendix A. Transport dynamics analysis for JFJ

We performed a dedicated set of simulations to characterise the atmospheric transport in backward LPDM simulations for JFJ as represented by different models in different configurations for 2009–2017. In order to analyse source sensitivity dependencies on domain size (Western Europe ("small") vs. Europe ("large")), LPDM implementation (FLEXPART vs. STILT) and meteorological input fields and associated spatial resolution (COSMO vs. ECMWF), we used four different combinations of these three parameters (Table A1). The simulations

are based on one assumed input field of idealized, positive $CO_2$ fluxes, which were kept constant in time and space for seven VTs based on the maps underlying the VPRM model. This analysis is designed to study atmospheric transport of chemically passive tracers released rather uniformly over the European continent to the high Alpine site and the obtained signals serve as a measure of PBL-influence of JFJ. It includes the total of the synthetic $CO_2$ concentration time-series from all seven VTs, alongside sub-groups comprising a) cropland, b) mixed forest, and c)

the total of the remaining 5 VTs. Studying the VT subgroups gives insight into the influence of spatial distributions of the sources within the domains under the given assumptions of uniform fluxes.

**Table A1.** Model combinations for transport dynamics analysis. E3 and E4 are the model configurations as used for the $CO_2$ concentration simulation in the main text.

| Ref. | LPDM | Weather Fields | Approximate Spatial Resolution (km$^2$) | Domain* | Integration period (d) | Release Height, (m asl) | Sampling Height (m) | Temporal Resolution |
|------|------|------|------|------|------|------|------|------|
| E1 | FLEXPART | ECMWF | 20×20 | EU | 10 | 3000 m | 100 | 3-hourly avgerage |
| E2 | FLEXPART | ECMWF | 20×20 | WEU | 10 (cropped) | 3000 m | 100 | 3-hourly avgerage |
| E3 | FLEXPART | COSMO7 | 7×7 | WEU | 4 | 3100 m | 50 | 3-hourly avgerage |
| E4 | STILT | ECMWF | 25×25 | EU | 10 | 3100 m | $0.5 \times h_{PBL}$ | snapshots every 3$^{rd}$ hour |

* "EU" and "WEU" refers to 33°N-73°N, -15-35°E, and 36.06-57.42°N, −11.92-21.04°E, respectively

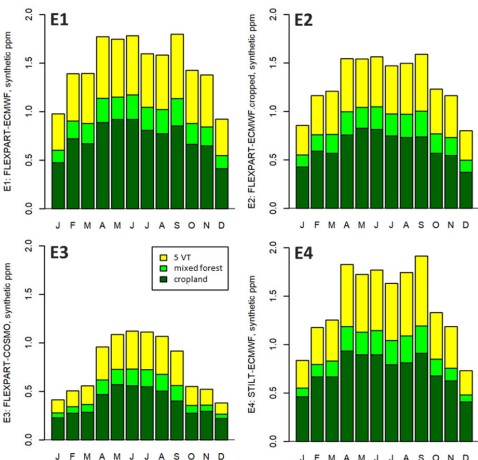

**Figure A1.** Mean monthly PBL-sensitivity (JFJ, 2009–2017) towards **i)** domain size (E1 vs. E2)**, ii)** meteorological input fields and spatial resolution (E2 vs. E3)**, iii)** LPDM implementation (E1 vs. E4)**, iv)** combinations (E3 vs. E4).





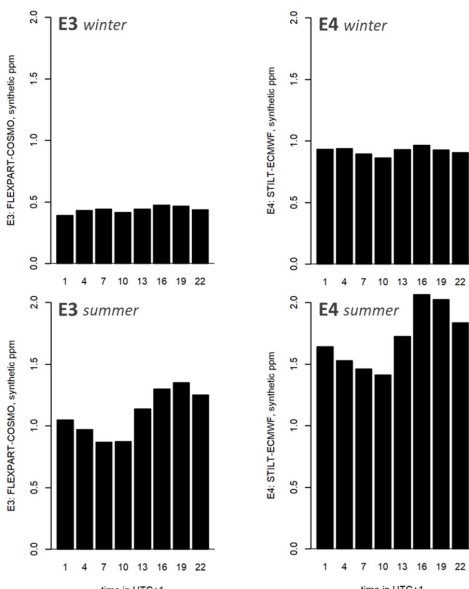


**Figure A2.** Mean diurnal PBL-sensitivity (JFJ, winter (DJF, top) and summer (JJA, bottom) for the period 2009–2017) for **a)** FLEXPART-COSMO ("E3") and **b)** STILT-ECMWF ("E4").


Figure A1 provides the multi-annual monthly means of the 3-hourly tracer concentrations at JFJ, and highlights the sensitivity towards domain size (E1 vs. E2), meteorological input fields and spatial resolution (E2 vs. E3), LPDM implementation (E1 vs. E4), and combinations of these (E3 vs. E4). Overall, we find that the synthetic $CO_2$ concentrations simulated at JFJ vary between the different models and configurations, as well as with seasonality and diurnal cycle. The analyses indicate a significant seasonality in the PBL-influence for all four configurations. Higher tracer concentrations are observed during the warm period (April-September) and relatively


lower tracer concentrations during the colder period (October-March). This confirms the generally stronger vertical transport during warm (and possibly sunny) days. Further, meteorological input fields and related spatial resolution (ECMWF vs. COSMO, i.e. E2 vs. E3) appear to have a larger influence compared to the LPDM implementation itself (FLEXPART vs. STILT, i.e. E1 vs. E4), and intensity discrepancies between the models used in the main text (E3, E4) are largest in winter, followed by summer, and smallest during transition periods. Concerning the domain


size, we find differences between different VT classes, which is owing to their heterogeneous spatial distribution as some VT classes are present predominantly inside (e.g. mixed forest) or outside (e.g. deciduous forests) the smaller domain boundaries; compare Figure S2. The smallest discrepancy was thus found for mixed forest (essentially 0 %), and a larger discrepancy (on average −15 %) was found for cropland, at the artificially assumed spatially and temporally constant fluxes. The influence of the LPDM implementation itself (FLEXPART vs. STILT, i.e. E1 vs.


E4) appears to be smaller than that of the meteorological fields and spatial resolution, generating differences mainly during winter periods, when FLEXPART-ECMWF yields a higher relative signal compared to STILT-ECMWF. In Figure A2, we present the PBL-influence on diurnal timescales, with up to 1.4 times higher synthetic $CO_2$ concentrations at JFJ during the afternoon and evening (maximum around 16:00–20:00 h, UTC+1) compared to the morning (minimum around 10:00 h, UTC+1). This is observed for FLEXPART-COSMO (E3) as well as STILT-


ECMWF (E4), and it is particularly pronounced during summer (June-August).





## Abbreviations and Definitions

| | |
|---|---|
| $f_b$ | $CO_2$ concentration in the background, expressed in ppm |
| $f_s$ | Regional contribution to the $CO_2$ concentration per category, expressed in ppm |
| $CO_2$.regional | Sum of all regional contributions to the $CO_2$ concentrations (fs) |
| $CO_2$.total | Sum of $CO_2$,regional and JCS-based $CO_2$ background ($f_b$) |
| $CO_2$.anthr | $CO_2$ concentration associated with all anthropogenic (anthr) categories |
| $CO_2$.cement | $CO_2$ concentration associated with cement production |
| $CO_2$.fuel | $CO_2$ concentration associated with all fuel categories |
| $CO_2$.gee | $CO_2$ concentration associated with gross ecosystem exchange (i.e. ecosystem uptake) (gee) |
| $CO_2$.nee | $CO_2$ concentration associated with net ecosystem exchange (nee) |
| $CO_2$.resp | $CO_2$ concentration associated with gross ecosystem respiration (resp) |
| $\delta^{13}C_a$ | $\delta^{13}$C-$CO_2$ estimate for atmospheric $CO_2$ at JFJ ‰ |
| $\delta^{13}C_b$ | $\delta^{13}$C-$CO_2$ estimate for the background $CO_2$, ‰ |
| $\delta^{13}C_m$ | $\delta^{13}$C-$CO_2$ mixed source signature for all $\delta^{13}C_s$ weighted with the $CO_2$ concentration ($f_s$), ‰ |
| $\delta^{13}C_s$ | $\delta^{13}$C-$CO_2$ source signature, ‰ |
| COSMO | Consortium for Small Scale Modelling |
| ECMWF | European Centre for Medium-Range Weather Forecasts |
| EDGAR | Emissions Database for Global Atmospheric Research |
| FLEXPART | Flexible Particle Model |
| JCS | Jena CarboScope based background estimate |
| LPDM | Lagrangian particle dispersion model |
| MACC-TNO | Monitoring Atmospheric Composition and Climate (provided by TNO) |
| QCLAS | Quantum Cascade Laser Absorption Spectrometer |
| STILT | Stochastic Time Inverted Lagrangian Transport |
| VPRM | Vegetation and Photosynthesis Respiration Model |

## Data & Code Availability

References to data/code are provide in main text/Supplement. Additional data will be made available online upon manuscript publication and further information may be requested from Lukas.Emmenegger@empa.ch.

## Author Contributions

SMP and SH wrote the manuscript with contributions from all authors. LE supervised the project. *Simulations*: UK, SMP, SH and DB prepared the annual scaling factors for the anthropogenic inventory, CG and TK prepared updated VPRM parameters. SH performed the $CO_2$ simulations with FLEXPART-COSMO. UK performed the $CO_2$ simulations with STILT-ECMWF. SH, DB, UK, CG, TK and SMP performed the transport dynamics analysis. *Observations*: BT, MST and LE provided the experimental data from QCLAS and CRDS. *Data Analysis*: SMP, SH, MST and DB prepared the data processing routines. SMP performed the model- and observation-based $\delta^{13}$C-$CO_2$ and $\delta^{13}C_m$ estimations, and overall data analyses and evaluations.

## Conflicting Interests

The authors declare that they have no conflict of interest.

## Acknowledgements

This research was supported by the Swiss National Science Foundation (ICOS-CH phase II, grant 20FI20_173691), the Swiss Federal Office for the Environment, the European Commission (RINGO, grant no. 730944), and the Global Atmosphere Watch Quality Assurance/Science Activity Centre Switzerland (QA/SAC-CH), funded by MeteoSwiss and Empa. SMP received funding from the Swiss National Science Foundation under project number P400P2_194390. We thank the International Foundation High Altitude Research Stations Jungfraujoch and Gornergrat for access to Jungfraujoch facilities and local support, and the Swiss National Supercomputing Centre (CSCS) under project ID s862 and the ICOS Carbon Portal for access to computational resources. We thank G. Janssens-Maenhout for providing the EDGAR v4.3 pre-release version, C. Rödenbeck for the Jena CarboScope Fields, TNO for the anthropogenic time-factors, U. Molteni for contributions to data analyses and graphical layout, and A. Jordan, H. Moossen and M. Rothe for providing the GC-FID and IRMS measurements (flask samples).



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
