# Peer review of "Analysis of regional CO2 contributions at the high Alpine observatory Jungfraujoch by means of atmospheric transport simulations and $\delta^{13}$ C"

_Atmospheric Chemistry and Physics, 2021_

## Referee Comment (RC1)

**Review of the manuscript "Analysis of regional CO$_2$ contributions at the high Alpine observatory Jungfraujoch by means of atmospheric transport simulations and δ$^{13}$C"**

General remark:

The data set presented in the manuscript is unique and valuable due to its long period, precise measurements and the remoteness of the station. Therefore, the presented rigorous analysis the data set is important. The analysis comprises measurements and model simulations, which are combined in a reasonable manner. The manuscript is well structured. Some of the passages are rather descriptive and could be refined. I recommend publication in ACP after some additional revisions concerning the following remarks:

Specific comments:

- Line 22: The authors comment that n R$^2$ of 0.4 is "remarkable" for the Alpine topography. However, a more objective description would rather be "acceptable" or "according to expectations".
- Line 69 – 85 and 114-125: This is much detail on other tracers, which are not investigated in the paper, and makes the paper lengthy and more diffuse. I recommend skipping or at least shortening discussion on CH4, CO, N2O, $^{18}$O-CO$_2$ and $^{14}$C as they are not the scope of the paper.
- Line 300: "And overall agreement within the extended compatibility parameters of the WMO". The extended goals reflect the less stringent requirements for urban and regional studies. However, as Jungfraujoch station is a remote station with only small influences of pollution, the authors should not reason with the extended WMO goals here. In the contrary, they should argue why not reaching the WMO goals for clean sites does still allow the analysis in this manuscript.
- Line 388: The authors describe the "fair-weather-effect", but they do not outline why this effect may be inadequately captured in the models.
- Line 395ff: As VPRM fluxes dominate the measured regional CO$_2$ signal (especially in summer), an estimation of uncertainty of VPRM fluxes would be very valuable. The authors make some qualitative statements about VPRM, but leave the reader without a clue on the uncertainty of these biogenic fluxes.
- Figure 6 and Line 529ff: The general pattern of δ$^{13}$C is captured by the models. However, small changes in δ$^{13}$C smaller than 0.1 ‰ may have a significant impact on the source signature. This is the case if the CO$_2$ discrepancy do not "match" the δ$^{13}$C discrepancy. The authors state that the discrepancies in CO$_2$ may contribute to the mismatch, but not why and to what degree. An analysis how the discrepancies could influence the source signature is needed. Especially as in Fig. 10 the measured source signature is compared to the model source signature.
- Table 4 and Figure 6: Instead of Table 4, an additional panel in Figure 6 showing the differences in modelled and observed δ$^{13}$C would be helpful. That way also phase differences and annual patterns would be visible.
- Fig 10 panel a: The authors derive the source signature by applying the moving Keeling plot method to their δ$^{13}$C and CO$_2$ measurements. They compare the source signature to the model to evaluate different filter criteria. However, it cannot be taken for granted that the model and measurements will show the same source signature. Especially, as no absolute agreement between measured and modelled δ$^{13}$C was achieved, any conclusion on filtering based on this comparison is not valid. A better test of filter criteria would be to apply the filters to the simulated δ$^{13}$C and CO$_2$ records to check if the model source signature can be obtained by the applied method.

- Line 783ff: The conclusion is missing a statement on how useful continuous isotope measurements actual are for the understanding of the carbon cycle at Jungfraujoch. In the manuscript, the authors use $\delta^{13}C$ qualitatively, but do not quantify biogenic or anthropogenic contributions. No significant additional information could be obtained from analyzing the continuous isotope measurements. I think the manuscript would benefit from a discussion on if and under which circumstances continuous $\delta^{13}C$ data can be useful for understanding the carbon cycle.
- Figure A1: The mean monthly discrepancy between E2 and E3 is very large. Ideally the authors would actually pinpoint the origin of this discrepancy by having an additional run with COSMO fields with a spatial resolution close to that of ECMWF.

Technical correction:

There are still some typos, comma and grammar mistakes in the manuscript.

---

## Referee Comment (RC2)

This work presents valuable data with a thorough analysis. The highresolution data over a long time period offers a great opportunity for studying CO2 in a remote location and the comparison of 2 atmospheric transport models is informative. Some of the text could be more concise to help with the flow, although generally the structure is logical and easy to follow. Publication of this article would make a valuable contribution to the community.

- Line 103: would benefit from a brief description of C3 & C4 plants e.g., examples of the types of plant in each category.
- Line 260: the explanation of the variables in equation 1 are a little hard to find as it is quite separated from the equation, reorganising this would make it easier to follow and refer back to.
- Section 2.2 previous studies have found that disaggregating the LPDM footprints back in time better captures the CO2 diurnal cycle as it does not assume that fluxes are constant for the duration of the simulation (i.e., 4 and 10 days) (e.g. (Denning et al., 1996; Gerbig et al., 2003; Gourdji et al., 2010, White et al., 2019). Do the authors expect that their results could be affected by this assumption?
- Figure 1 are the STILT-ECMWF peaks larger because of the PBL contribution discussed in section 3.1.1 A) if so a reference to this figure in section A) would help visualise the effect; if not does the author have some idea of what causes this?
- Lines 480-490 the authors suggest that some of the difference between the results from the 2 LPDM runs are due to difference in resolution, have the authors attempted to regrid the FLEXPART-COSMO to the STILT-ECMWF resolution to check how that affects the performance and if it can account for the difference as suggested?
- Figures 5 and 8 should be much bigger to make out the labels and detail
- Figure 6 it would be useful to include a zoom in on specific time periods as it is difficult to pick out how well the model matches the observations e.g., when discussing mismatch throughout the summer
- Figure 9 what is meant by 'lumps of ecosystem'?

---

## Author Comment (AC1)

**AUTHOR COMMENT (Pieber et al., ACPD, 2021)**

We thank both referees for taking the time to evaluate our manuscript. This document summarizes our replies and modifications to address the referees' comments. To facilitate the process, all referee comments are copied here in grey font, and our replies (marked as author comment, AC) are provided in blue font. Selected revision of the manuscript text are indicated here in *brown italic font* while *black italic font* indicates current manuscript text. All modifications made during the revision are available in the "track change" version of the revised manuscript.

**REFEREE 1 (R1)**

**R1 General remark:**
The data set presented in the manuscript is unique and valuable due to its long period, precise measurements and the remoteness of the station. Therefore, the presented rigorous analysis the data set is important. The analysis comprises measurements and model simulations, which are combined in a reasonable manner. The manuscript is well structured. Some of the passages are rather descriptive and could be refined. I recommend publication in ACP after some additional revisions concerning the following remarks.

AC We thank the referee for their time and availability to assess our manuscript and have addressed the specific comments and questions in the following.

**R1 Specific comments:**
**R1.1** - Line 22: The authors comment that an $R^2$ of 0.4 is "remarkable" for the Alpine topography. However, a more objective description would rather be "acceptable" or "according to expectations".

AC The word "remarkable" was rephrased to read "according to expectations" or removed in all related sections.

**R1.2** - Line 69 – 85 and 114-125: This is much detail on other tracers, which are not investigated in the paper, and makes the paper lengthy and more diffuse. I recommend skipping or at least shortening discussion on $CH_4$, CO, $N_2O$, $^{18}O$-$CO_2$ and $^{14}C$ as they are not the scope of the paper.

AC We believe that our extended introduction allows readers who are not yet familiar with the challenges of disentangling sources and sinks of GHGs in the atmosphere, in particular at remote locations, to obtain a broader overview. This is particularly true regarding the challenges of disentangling ecosystem and anthropogenic $CO_2$, and regarding the limitations of single tracer approaches in such context. We believe that it is important to point out to those not so familiar with stable isotopes, why, for instance, $^{18}O$-$CO_2$, although observations are available at JFJ, was not addressed in our manuscript in a double-isotope approach. We also believe it is important to highlight, that $^{13}C$-$CO_2$ would best be used for a separation between different anthropogenic fuel types, after a separation between the anthropogenic or ecosystem contributions has already been accomplished using e.g., $^{14}C$-$CO_2$. We agree that some of this information may be better suited for the final discussion that puts our findings into an overall context; yet, we wish to allow the reader to be informed about the potential and limitations of sole $\delta^{13}C$-$CO_2$ analysis at JFJ right from the very beginning of the manuscript. Thus we have opted for only minor adjustments of this section.

**R1.3** - Line 300: "And overall agreement within the extended compatibility parameters of the WMO". The extended goals reflect the less stringent requirements for urban and regional studies. However, as Jungfraujoch station is a remote station with only small influences of pollution, the authors should not reason with the extended WMO goals here. In the contrary, they should argue why not reaching the WMO goals for clean sites does still allow the analysis in this manuscript.

AC We agree with the referee. We have removed the reference to WMO and the section was updated as follows:
*"A comparison of flask sample measurements with the QCLAS measurements for 2009–2017 indicates very good agreement, typically within ±0.2 ppm for $CO_2$ and ±0.1 ‰ for $\delta^{13}C$-$CO_2$, as well as no apparent systematic bias as function of time or signal intensity. It should be noted that the data and sample collection for in-situ measurements (QCLAS) and offline samples (flasks) was not primarily designed to assess an inter-comparison between the two measurements systems. In particular, uncertainties exist regarding the accurate matching of time stamps. Therefore, the real agreement of the measurements is likely even better."*

**R1.4** - Line 388: The authors describe the "fair-weather-effect", but they do not outline why this effect may be inadequately captured in the models.

AC The fair-weather effect is driven by combination of day-time $CO_2$ uptake in the PBL and increased vertical export of PBL air to JFJ through thermally-induced flow systems in complex terrain (up-slope, up-valley, see e.g., Rotach et al., 2014 (DOI: 10.1175/BAMS-D-13-00109.1). In order for the transport models to adequately account

for thermally induced flow systems they would need to resolve the important topographic features, which is not completely the case for the two model systems employed here. Neither at 0.25°× 0.25° (STILT) nor at 7 km × 7 km (FLEXPART) horizontal resolution it can be expected that vertical export from the PBL in complex terrain is correctly represented. We expanded our previous statement to: *"Such "fair-weather effects" may be inadequately captured in the models, as the vertical export of PBL air in these situations is driven by thermally-induced flow systems in complex terrain (up-slope, up-valley, see Rotach et al., 2014) that cannot be adequately resolved at the present model horizontal resolution."*

**R1.5** - Line 395ff: As VPRM fluxes dominate the measured regional CO2 signal (especially in summer), an estimation of uncertainty of VPRM fluxes would be very valuable. The authors make some qualitative statements about VPRM, but leave the reader without a clue on the uncertainty of these biogenic fluxes.

**AC** An assessment of uncertainties in daily fluxes is estimated in Kountouris et al. (2015) (DOI: 10.5194/bg-12-7403-2015), based on a comparison with eddy covariance flux observations, to be 2.5 µmol m$^{-2}$ s$^{-1}$ for VPRM, and typical spatial error correlation of around 100 km corr. length and a temporal correlation of 30 days. This uncertainty estimate is between eddy covariance data and simulations using VPRM. To estimate the impact of this uncertainty on the simulated $CO_2$, however, full propagation of the error would be needed, including spatial and temporal correlation. As VPRM is used in many inversion studies, the corresponding error in simulated $CO_2$ can alternatively be assessed based on the change from prior to posterior model-data mismatch. Based on Table 3 in the technical note of Kountouris et al. (2018) (DOI: 10.5194/acp-18-3027-2018), typical numbers for mountain sites such as JFJ are around 4 ppm (prior), which drop to about 1.5 ppm for posterior fluxes (the assumed model-data mismatch error). We have added this information in the revised manuscript.

**R1.6** - Figure 6 and Line 529ff: The general pattern of $\delta^{13}C$ is captured by the models. However, small changes in $\delta^{13}C$ smaller than 0.1 ‰ may have a significant impact on the source signature. This is the case if the $CO_2$ discrepancy do not "match" the $\delta^{13}C$ discrepancy. The authors state that the discrepancies in $CO_2$ may contribute to the mismatch, but not why and to what degree. An analysis how the discrepancies could influence the source signature is needed. Especially as in Fig. 10 the measured source signature is compared to the model source signature.

**AC**
- Regarding Figure 6: The proposed analysis is provided already in Section 3.2.3 C (which is now updated to 3.2.2 C and D) and Table 5 as well as the associated content in the SI. Section 3.2.3 B (updated to 3.2.2 B) discusses aspects related to the background estimation and was further extended during the revision; in addition, Section 3.2.3 A (updated to 3.2.2 A) already discusses selected scenarios regarding the assumptions of source signatures, most prominently, the contribution of C4 plants. *"Replacing 7 % of the C3-related $CO_2$ with C4-related $CO_2$ would marginally change the source signature of crops (< 1 ‰, and that of the overall ecosystem signal by even less); however, generally $\delta^{13}C_m$ would become more enriched and thus the discrepancy between model and observations larger. Reducing a potential C4-related $CO_2$ fraction instead would make $\delta^{13}C_m$ less enriched and thus bring the simulations data into slightly better agreement with observations at JFJ. Indeed, the ecosystem assumptions for the Northern Hemisphere are based on data collected in the USA and might be characterised by a higher C4 fraction than the footprint region for JFJ."*
- Regarding Figure 10 (updated to figure number 11) - please see our reply to R1.8; the text mentioning a direct comparison of observation-based ("measured") and model-based $\delta^{13}C_m$ has been revised.

**R1.7** - Table 4 and Figure 6: Instead of Table 4, an additional panel in Figure 6 showing the differences in modelled and observed $\delta^{13}C$ would be helpful. That way also phase differences and annual patterns would be visible.

**AC** An additional figure (new Figure 7) has been added in 3.2.1 and parts of section 3.2.1 were moved to 3.2.2 D; alongside the presentation of the additional figure (new Figure 7), we have extended the sections discussing the assumptions for $\delta^{13}C_b$ and updated any related data, figures and text as needed (see track change version).

**R1.8** - Fig 10 panel a: The authors derive the source signature by applying the moving Keeling plot method to their $\delta^{13}C$ and $CO_2$ measurements. They compare the source signature to the model to evaluate different filter criteria. However, it cannot be taken for granted that the model and measurements will show the same source signature. Especially, as no absolute agreement between measured and modelled $\delta^{13}C$ was achieved, any conclusion on filtering based on this comparison is not valid. A better test of filter criteria would be to apply the filters to the simulated $\delta^{13}C$ and $CO_2$ records to check if the model source signature can be obtained by the applied method.

**AC** Indeed, it cannot be taken for granted that the model and measurements will show the same source signature. Obtaining the modelled $\delta^{13}C_s$ signature is dependent on a correct simulation of $CO_2$ quantities (that agrees with $CO_2$ observations). Obtaining a "correct" Keeling-plot-derived source signature is attainable only under stringent conditions described in section 3.2.4 (section was updated to number "3.2.3" during revision). Thus, both approaches have their particular limitations and are independent from each other. An assessment as suggested by the referee has already been presented by Vardag et al. in 2016 (DOI: 10.5194/bg-13-4237-2016) and it was not our

intention to repeat this exercise in our manuscript. As the model time resolution is 3-hourly, such assessment would require using the 3-hourly observations data set instead of the 1-hourly averaged data for the Keeling-plot method. This would result in decreased $\Delta CO_2$ and limit the Keeling-plot method. Any filter criteria derived like this might not be directly transferrable to the hourly or 10-min data set, and therefore such additional assessment on a 3-hourly data set would be of limited meaning. Instead, we evaluate the applied filtering criteria to the observation-based data not by comparing with the model-based data, but by assessing whether the observation-based values are physically meaningful or not (in particular, whether any unreasonable values greater than 0‰ are generated); this approach is more suitable for the scope of our study. We have slightly revised the relevant sections, to make our points more clear. In particular, we have added the following lines in section 3.2.4 (updated to 3.2.3 during revision): *"Note, that we do not expect that model-based $\delta^{13}C_m$ and observation-based $\delta^{13}C_m$ can be compared directly with each other, as the model-based $\delta^{13}C_m$ are calculated for 3-hourly resolution and, most importantly, are not restricted to situations when the underlying $CO_2$ simulations match the $CO_2$ observations."*

**R1.9** - Line 783ff: The conclusion is missing a statement on how useful continuous isotope measurements actual are for the understanding of the carbon cycle at Jungfraujoch. In the manuscript, the authors use $\delta^{13}C$ qualitatively, but do not quantify biogenic or anthropogenic contributions. No significant additional information could be obtained from analyzing the continuous isotope measurements. I think the manuscript would benefit from a discussion on if and under which circumstances continuous $\delta^{13}C$ data can be useful for understanding the carbon cycle.

**AC** The referee is correct in stating that we have not used the continuous $\delta^{13}C$ observations for a quantitative $CO_2$ source attribution at JFJ (i.e., we have not quantitatively disentangled regional contributions of the multi-source and broadly distributed atmospheric $CO_2$ at a remote site). Instead, our manuscript focuses first on novel simulations of regional $CO_2$ at JFJ with two different models in comparison to highly time-resolved $CO_2$ observations. Those simulations, which capture the $CO_2$ observations to a satisfactory degree, confirm our expectations that regional $CO_2$ contributions at JFJ are highly mixed and of both, ecosystem and anthropogenic origin. Consequently, it is expected and unlikely that $\delta^{13}C$ alone will be sufficient to quantify biogenic and anthropogenic contributions – as we have extensively reviewed in the introduction (see also R1.2). Consequently, considering the large number of degrees of freedom in the system we did not make a firm quantitative conclusion regarding the $CO_2$ composition based on $\delta^{13}C$. Instead, we discuss the $\delta^{13}C$ measurements in a qualitative sense, by providing a comparison between a forward-simulation of $\delta^{13}C$ and our continuous observations. This consequently enables the analysis presented in section 3.2.3 C, which indicates that the mismatch between $\delta^{13}C$ simulations and observations is related in addition to the underlying assumptions for $\delta^{13}C$ source signature and background also to uncertainties in the simulated $CO_2$ quantities. In addition, we provide observation-based source signatures derived from a Keeling-plot approach; the obtained data set, however, is small. Yet, the section 3.2.4 (updated to 3.2.3) highlights the need of continuous $\delta^{13}C$ observations in the determination of such source signatures, e.g. in the following statement: *"The high-precision of the $\delta^{13}C$-$CO_2$ measurements and the high time-resolution available from the QCLAS instrument allow to compensate the low $\Delta CO_2$ and to limit fitting uncertainties....").* The manuscript concluded the section 3.2.4 (updated to 3.2.3) with the following information, which in our opinion addresses some of the points raised by the referee. This paragraph has thus been updated and moved to the conclusions: *"A further disaggregation of observation-based $\delta^{13}C_m$ using mass balance approaches and assumptions for the end-members in order to learn more about the $CO_2$ regional composition for any further comparison to the simulated $CO_2$ regional composition, was not attempted here, given the small number of observation-based $\delta^{13}C_m$ values obtained. This may be the focus in future studies. However, we expect that it will remain challenging to disentangle fuel and ecosystem respiration signals from observation-based $\delta^{13}C_m$ alone, considering that the simulated regional $CO_2$ fractions at JFJ indicate approximately equal amounts even during the winter. In addition, the solid and liquid fuel emissions $\delta^{13}C_s$ end-member assumptions overlap with C3 plant respiration signatures. Thus, while such $\delta^{13}C_s$ source apportionment approaches prove meaningful among either the anthropogenic or the ecosystem carbon pool, they are of more limited use as a singular tracer when the carbon pools are mixed."* We have revised the conclusions further as presented in the track change document to address the referees comment further.

**R1.10** - Figure A1: The mean monthly discrepancy between E2 and E3 is very large. Ideally the authors would actually pinpoint the origin of this discrepancy by having an additional run with COSMO fields with a spatial resolution close to that of ECMWF.

**AC** Whether the large differences between E2 and E3 result from model resolution alone or being inherent to the model system (COSMO vs. ECMWF-IFS) cannot be answered here. Performing additional COSMO simulations at a similar spatial resolution, as the applied ECMWF-IFS data, would be computationally expensive and beyond the scope of this publication. Note, that both NWP data sets (COSMO and ECMWF-IFS) are operational products of MeteoSwiss and ECMWF, respectively, and not part of our own analysis. They are merely used as input data for FLEXPART and STILT. From previous studies, we know that differences between FLEXPART-COSMO and FLEXPART-IFS (same resolution as here) for sites in less complex terrain were smaller as seen here for Jungfraujoch. Although, this suggests that resolution may be the main influence, we deem this statement too speculative to include it in the manuscript.

**R1 Technical correction:** There are still some typos, comma and grammar mistakes in the manuscript.
**AC** We have corrected typos, comma and grammar mistakes.

**REFEREE 2 (R2)**

**R2 General remark:**
This work presents valuable data with a thorough analysis. The high resolution data over a long time period offers a great opportunity for studying $CO_2$ in a remote location and the comparison of 2 atmospheric transport models is informative. Some of the text could be more concise to help with
the flow, although generally the structure is logical and easy to follow. Publication of this article would make a valuable contribution to the community.
**AC** We thank referee 2 for their time and availability to assess our manuscript. The specific comments and questions are addressed in the following and have additionally revised the text for redundancies.

**R2 Specific comments:**
**R2.1** Line 103: would benefit from a brief description of C3 & C4 plants e.g., examples of the types of plant in each category.
**AC** Only few terrestrial species (around 3%) follow the C4 carbon fixation pathway, and they fall primarily in the category of grasses (e.g., clover), although some selected crops (maize, sugar cane, sorghum and various kinds of millet), and few trees and desert shrubs follow this photosynthetic pathway as well. Globally, C3 carbon fixation dominates the biosphere. We have added an additional note on plants that follow the C3 and C4 plants, although the introduction and discussion already mentioned the minor relevance of C4 plants other than maize to our study.

**R2.2** Line 260: the explanation of the variables in equation 1 are a little hard to find as it is quite separated from the equation, reorganising this would make it easier to follow and refer back to.
**AC** We agree and will consider this during final type setting.

**R2.3** Section 2.2 - previous studies have found that disaggregating the LPDM footprints back in time better captures the CO2 diurnal cycle as it does not assume that fluxes are constant for the duration of the simulation (i.e., 4 and 10 days) (e.g. (Denning et al., 1996; Gerbig et al., 2003; Gourdji et al., 2010, White et al., 2019). Do the authors expect that their results could be affected by this assumption?
**AC**
- FLEXPART-COSMO: We did not use 'total' footprints when calculating the regional concentration increments but time-resolve (disaggregated) footprints with time step of 3 hours. For each of these 3-hourly intervals both biogenic and anthropogenic $CO_2$ fluxes were varied according to VPRM and typical time functions by sector. This procedure was mentioned already in the manuscript (see Section 2.2.1 and 2.3.1).
- This is true in a similar manner for STILT-ECMWF, although this information was not included in the manuscript initially. It is added in the revised version for clarification. In STILT-ECMWF we typically use hourly time resolution for coupling transport with fluxes.

**R2.4** Figure 1 - are the STILT-ECMWF peaks larger because of the PBL contribution discussed in section 3.1.1 A) - if so a reference to this figure in section A) would help visualise the effect; if not does the author have some idea of what causes this?
**AC** Yes, this is indeed part our interpretation, aside of further aspects such as the domain size mentioned in Section 3.1.1 A and in particular in the analysis presented in Appendix A1. We have added a reference and note in the text as suggested in both, Section 3.1.1 A as well as in Appendix A1.

**R2.5** Lines 480-490 – the authors suggest that some of the difference between the results from the 2 LPDM runs are due to difference in resolution, have the authors attempted to regrid the FLEXPART-COSMO to the STILTECMWF resolution to check how that affects the performance and if it can account for the difference as suggested?
**AC** The only way this could be done correctly would be running COSMO simulations at a similar resolution as the one of the ECMWF data. This would be computationally too demanding for the scope of this publication (see also reply to reviewer 1) and has not been done. Alternatively, the resolution of the COSMO-7 data could be artificially reduced. Once again, this is a non-trivial task since mass consistency of the flow field would need to be considered. Furthermore, it is not obvious that reducing the output resolution of a high-resolution model run would yield similar results to running the model at coarse resolution.

**R2.6** Figures 5 and 8 – should be much bigger to make out the labels and detail
**AC** We agree and will consider this for the final type setting. We have also added a full page version of both Figure 5 and Figure 8 (new figure number is 9) in the SI.

**R2.7** Figure 6 - it would be useful to include a zoom in on specific time periods as it is difficult to pick out how well the model matches the observations e.g., when discussing mismatch throughout the summer
AC Zoom version for 2012, 2013, 2014, 2015 were added in the SI. See also modifications regarding R1.7.

**R2.8** Figure 9 - what is meant by 'lumps of ecosystem'?
AC Thank you for pointing this out. We have deleted "different lumps of" in the Figure 9 caption (=new figure number is 10), as it was confusing. The figure caption now reads as follows: *"Time series **a)** model-based $\delta^{13}C_m$ (Eq. (1)), **b-c)** model-based $\delta^{13}C_m$ for ecosystem-, fuel- and cement-related $CO_2$"*

**REFERENCES**

- Kountouris, P., Gerbig, C., Rödenbeck, C., Karstens, U., Frank Koch, T. and Heimann, M.: Technical Note: Atmospheric CO2 inversions on the mesoscale using data-driven prior uncertainties: Methodology and system evaluation, Atmos. Chem. Phys., 18(4), 3027–3045, doi:10.5194/acp-18-3027-2018, 2018b.
- Kountouris, P., Gerbig, C., Totsche, K. U., Dolman, A. J., A. Meesters, A. G. C., Broquet, G., Maignan, F., Gioli, B., Montagnani, L. and Helfter, C.: An objective prior error quantification for regional atmospheric inverse applications, Biogeosciences, 12(24), 7403–7421, doi:10.5194/bg-12-7403-2015, 2015.
- Rotach, M. W., Wohlfahrt, G., Hansel, A., Reif, M., Wagner, J. and Gohm, A.: The world is not flat: Implications for the global carbon balance, Bull. Am. Meteorol. Soc., 95(7), 1021–1028, doi:10.1175/BAMS-D-13-00109.1, 2014.
- Vardag, S. N., Hammer, S. and Levin, I.: Evaluation of 4 years of continuous δ13C(CO2) data using a moving Keeling plot method, Biogeosciences, 13(14), 4237–4251, doi:10.5194/bg-13-4237-2016, 2016.